

# Diurnal cycle of the semi–direct effect over marine stratocumulus in large–eddy simulations

Ross J. Herbert[1], Nicolas Bellouin[1], Ellie J. Highwood[1], Adrian A. Hill[2]

[1]Department of Meteorology, University of Reading, Reading, RG6 6BB, UK
[2]Met Office, Fitzroy Road, Exeter, EX1 3PB, UK

*Correspondence to*: Ross Herbert (r.j.herbert@reading.ac.uk)

The rapid adjustment, or semi–direct effect, of marine stratocumulus clouds to elevated layers of absorbing aerosols may enhance or dampen the radiative effect of aerosol–radiation interactions. Here we use large eddy simulations to investigate the sensitivity of stratocumulus clouds to the properties of an absorbing aerosol layer located above the inversion layer. The sign of the daily mean semi–direct effect depends on the properties of the aerosol layer, the properties of the boundary layer, and the model setup. Diurnal variations in the cloud response mean that an instantaneous semi–direct effect is unrepresentative of the daily mean, and that observational studies may under– or over–estimate semi–direct effects depending on the observed time of day. The observed role of the distance between the cloud top and the absorbing layer in modulating the strength of the cloud and radiative response is reproduced by the large eddy simulations. Both cloud response and semi–direct effect increase for thinner, denser, layers of absorbing aerosol located nearer the cloud layer. The cloud response is particularly sensitive to the mixing state of the boundary layer: well-mixed boundary layers generally result in a negative daily mean semi–direct effect, and poorly mixed boundary layers result in a positive daily mean semi–direct effect. Properties of the boundary layer and model setup, particularly the sea surface temperature, precipitation, and properties of the air entrained from the free troposphere, also impact the magnitude of the semi–direct effect and the timescale of adjustment. These results suggest that the semi–direct effect simulated by coarse-resolution models may be erroneous because the cloud response is sensitive to small-scale processes, especially the sources and sinks of buoyancy.

## 1    Introduction

Semi-permanent decks of marine stratocumulus clouds represent an important negative radiative effect within the Earth's energy budget (Hartmann et al., 1992; Hartmann and Short, 1980; Wood, 2012). In addition, the sharp inversion layer and small–scale turbulent processes that characterise the formation and maintenance of these clouds represent considerable uncertainty in climate models, so stratocumulus clouds remain a key uncertainty in future climate projections (Bony and Dufresne, 2005; Klein et al., 2017; Wood, 2012). Marine stratocumulus clouds are sensitive to sea surface temperature (SST) and large–scale atmospheric properties both above the inversion, like subsidence rate and thermodynamic properties of the





overlying air mass, and below the inversion, like cloud condensation nuclei sinks and sources, that impact turbulent processes
and dynamics throughout the boundary layer (e.g., Bretherton et al., 2013; Feingold et al., 2010; Sandu et al., 2010). Therefore,
small changes to these properties could result in large changes to the fluxes of radiation in the atmosphere.
Perturbations to the aerosol distribution result in a radiative forcing through both aerosol–radiation and aerosol–cloud
interactions; this distinction separates the radiative forcing caused by aerosol scattering and absorption of longwave and
shortwave radiation from that caused by the availability of cloud condensation nuclei. Aerosol–cloud interactions lead to
changes in cloud albedo and subsequent rapid adjustments to the cloud properties that include changes to precipitation and
cloud evolution (Sherwood et al., 2015). Aerosol–radiation interactions result in instantaneous changes to the extinction profile
(also referred to as the direct radiative effect) and therefore heating profile, which lead to rapid adjustments in the physical and
radiative properties of the cloud (referred to in this paper as the semi–direct effect, SDE, for convenience). Quantifying rapid
adjustments is important as they may act to dampen or strengthen the instantaneous forcing. Aerosol–radiation interactions
represent an important uncertainty in the anthropogenic radiative forcing of the climate over the industrial era, especially from
absorbing aerosol species such as black carbon which may result in pronounced semi–direct effects (Boucher et al., 2013). In
a recent climate model intercomparison study Stjern et al. (2017) found that a ten–fold increase in black carbon emissions
resulted in a strong positive direct effect which was partially offset by a negative SDE. Although all models agree on the sign
(negative) they disagree on the size of that offset, from 12 to 63 % for the models studied by Stjern et al. (2017). High–
resolution models that can sufficiently represent the dominant processes within the boundary layer and cloud are a powerful
benchmark to test the realism of the response simulated by the climate–scale models.

During the African dry season, which lasts from August to October, plumes of strongly absorbing biomass burning aerosol
from central Africa are transported westward over the semi–permanent marine stratocumulus deck of the Southeast Atlantic
Ocean, where they eventually subside and mix into the boundary layer (Das et al., 2017). Observational and modelling studies
suggest that elevated absorbing layers result in thicker clouds and a negative SDE (Adebiyi and Zuidema, 2018; Johnson et
al., 2004; Wilcox, 2010), and may impact the stratocumulus–to–cumulus transition process (Yamaguchi et al., 2015; Zhou et
al., 2017). Once mixed into the cloud layer the absorbing aerosol exerts aerosol–radiation interactions that enhance cloud
evaporation (Hill and Dobbie, 2008; Johnson et al., 2004) and aerosol–cloud interactions that impact microphysical and
dynamical processes (e.g., Feingold et al., 2010; Gordon et al., 2018; Hill et al., 2009). Observational studies have used satellite
retrievals from the NASA A–Train to investigate the interaction between clouds and absorbing aerosol over the Southeast
Atlantic. Wilcox (2010) used co–located CALIPSO, OMI, and AMSR–E retrievals and found that for all overcast scenes liquid
water path (LWP) increased for high aerosol loading. This response was attributed to absorbing aerosol layers above the cloud
top enhancing the heating rate and decreasing entrainment across the inversion. However, satellites do not provide direct
observations of entrainment and an alternative explanation could be that the aerosol layers travel in relatively moist layers
(Adebiyi et al., 2015), increasing moisture transport across the inversion layer, even if entrainment remained unchanged. In a



study with a similar methodology, Costantino and Bréon (2013) separated the CALIPSO–derived aerosol layer heights into
cases when the smoke was close to (< 100 m) and well–separated (< 750 m) from the cloud top. The authors found that when
the aerosol layers are well separated from cloud top the LWP and cloud optical thickness showed no statistically significant
dependence on aerosol loading. These results are supported by Adebiyi and Zuidema (2018) who used satellite observations
and reanalysis products to show evidence that the sensitivity of low–cloud cover to elevated aerosol layers increased for small
cloud–aerosol gaps. These observations suggest that the distance between the elevated aerosol layer and cloud layer plays an
important role in the strength of the SDE. Additionally, a recent satellite study of cloud–aerosol gaps by Rajapakshe et al.
(2017) suggests that the elevated aerosol layers may be closer to the cloud than previously thought, which demonstrates that
elevated layers may have an even more important impact on the clouds.

The observations hint at the potential importance of the extent of cloud–aerosol gap for the SDE. However, this complexity is
not reflected in the frameworks presented in current reviews (Bond et al., 2013; Koch and Del Genio, 2010), and there is a
lack of high–resolution modelling studies investigating the SDE from elevated layers of absorbing aerosol. Johnson et al.
(2004) used large–eddy simulation (LES) to investigate the semi–direct of absorbing aerosols on non-precipitating marine
stratocumulus. In an experiment where a ~1 km thick layer of absorbing aerosol, with an aerosol optical depth (AOD) of 0.2
at 550 nm, was present above the marine boundary layer throughout a 48–hr simulation, the absorbing aerosol enhanced the
temperature inversion at the top of the boundary layer, weakening the entrainment rate across the inversion, and producing a
shallower, moister boundary layer and a higher LWP. The 48–hr mean SDE was estimated to be -9.5 $Wm^{-2}$, almost entirely
cancelling a direct effect of +10.2 $Wm^{-2}$. Yamaguchi et al. (2015) and Zhou et al. (2017) used LES models to investigate the
transition of marine stratocumulus to cumulus in the presence of a smoke layer. As the marine boundary layer deepened, the
cloud–aerosol gap decreased until the smoke layer made contact with the cloud layer. Both studies found little LWP response
when the smoke layer was separated by a no–aerosol gap. Yamaguchi et al. (2015) found that the elevated smoke layer reduced
boundary layer turbulence and cloud cover through a decrease in longwave cloud–top cooling. By isolating the aerosol heating
above and below the boundary layer top Zhou et al. (2017) found that when the layer was directly above the inversion layer
the elevated aerosol layer strengthened the inversion, inhibiting entrainment, and increased LWP and cloud cover, resulting in
a negative SDE. Global models have also been used to investigate the radiative impact of biomass burning aerosol in
stratocumulus regions (e.g., Lu et al., 2018; Penner et al., 2003; Sakaeda et al., 2011), however, Das et al. (2017) show that
these coarser resolution models may be unable to reproduce the observed vertical distribution of absorbing aerosol layers over
the southeast Atlantic, resulting in an under–representation of elevated aerosol layers and increased uncertainty in their
radiative impact.

In summary, observation and modelling studies suggest that the diurnal cycle and evolution of marine stratocumulus are
strongly impacted by the presence of absorbing aerosol layers at and above the top of the boundary layer. The SDE may act to
counteract or enhance the direct effect, resulting in either a small or large net radiative effect from aerosol–radiation


interactions. Yet the sensitivity of the SDE to the properties of the elevated aerosol layer has not been fully investigated. In
this study the UK Met Office Large Eddy Model (LEM) is used to investigate and quantify the impact that the properties of
an elevated absorbing aerosol layer have on the cloud and radiative response of marine stratocumulus. Section 2 presents the
LEM and its configuration and introduces a set of experiments designed to assess the SDE and its sensitivity to the aerosol
layer properties. Section 3 focuses on a single experiment to understand the processes that drive the cloud response and SDE,
then assesses the sensitivity of this response to the aerosol layer properties. Section 3 also investigates the robustness of that
assessment to the processes that affect the maintenance of the cloud, namely precipitation, sea surface temperature, and
boundary layer depth. Section 4 summarises the results, comparing to other modelling studies and observations, and discussing
the limitations of this study and identifying remaining questions.
**2    Model description and setup**
**2.1    Description of model**
The LEM (Gray et al., 2001) is a non–hydrostatic high–resolution numerical model that explicitly resolves the large–scale
turbulent motions responsible for the energy transport and flow. Sub–grid scale turbulence responsible for the dissipation of
kinetic energy is parameterised. Prognostic variables are the three–dimensional velocity fields $(u,v,w)$, liquid–water potential
temperature $(\theta_l)$, and mass–mixing ratios of water vapour $(q_v)$, liquid water $(q_l)$, and rain $(q_r)$. Liquid water mass is prognosed
at every grid point using a condensation–evaporation scheme in which excess supersaturation is converted to liquid water and
vice versa for sub–saturated air. Warm rain processes are represented by a single–moment microphysics scheme that includes
autoconversion and cloud droplet collection following Lee (1989), sedimentation of rain, and evaporation of rain into dry air.
The influence of aerosol on cloud droplet number concentration is not included in this study and cloud droplet number is fixed
to 240 cm$^{-3}$ for microphysical processes. Surface fluxes are calculated using Monin–Obukhov similarity theory (Monin and
Obukhov, 1954) with a prescribed constant sea surface temperature. A damping layer that relaxes all prognostic variables to
their horizontal mean is present above an altitude of 775 m (~150 m above the cloud layer; see Sect. 2.2) with a height scale
of 650 m and a timescale of 30 s. This prevents the reflection of gravity waves at the rigid top boundary and prevents the
production of trapped buoyancy waves above the inversion layer (Ackerman et al., 2009). The subsidence rate $w_s$ is represented
by a height dependent function $w_s(z) = -Dz$ for which large–scale divergence $(D)$ is prescribed. The model is run with a variable
time step with a maximum of 0.5 seconds. The LEM radiation scheme, described by Edwards and Slingo (1996), is a two–
stream solver with six shortwave spectral bands and eight longwave bands that calculates the vertical distribution of radiative
fluxes and heating rates. The scheme includes six aerosol species with wavelength and humidity–dependent mass absorption
coefficients, mass scattering coefficients, and asymmetry factors. A single value for the mean cloud droplet effective radius of
10 μm is prescribed in the radiation scheme.



**2.2    Model setup**
All simulations are three dimensional. The model domain is 5200 m in the horizontal with a horizontal grid resolution of 40 m,
and 2600 m in the vertical with a variable vertical grid resolution with ~6 m resolution at the cloud top and inversion and less
than 10 m throughout the boundary layer (BL). The LEM is configured here to produce a stratocumulus with a consistent
diurnal cycle over an 8–day timescale. The initial profiles of $\theta_l$ and $q_t$ were taken from Johnson et al. (2004) and based on
subtropical marine stratocumulus observations from the First International Satellite Cloud Climatology Project Regional
Experiment (FIRE) (Hignett, 1991) in the subtropical Pacific Ocean. A series of 10–day simulations without absorbing aerosol
were run with varying subsidence rates to obtain steady–state profiles of $\theta_l$ and $q_t$ that would produce a consistent stratocumulus
layer with a maximum cloud top height of 600 m. The resulting initialisation profiles are shown in Table 1; the BL is 0.6 g kg$^{-1}$
drier than in Johnson et al. (2004) and Hill et al. (2008) due to the inclusion of precipitation in our study and a cooler SST,
which was necessary in order to attain a similar cloud LWP to these studies. The large–scale divergence $D$ is set to $5.5 \times 10^{-6}$ s$^{-1}$,
giving a subsidence rate of $w_s$ = -3.3 mm s$^{-1}$ at the cloud top. $D$ and $w_s$ are within the observed range for marine stratocumulus
regions (Zhang et al., 2009) and of similar magnitude to other stratocumulus LES studies (e.g., Johnson et al., 2004; De Roode
et al., 2014). The initial profiles describe a well–mixed moist BL capped by a sharp (10 K) inversion at 600 m with a warm
and dry free troposphere (FT) above the inversion. To account for a source of large–scale heat divergence a cooling rate of
0.1 K day$^{-1}$ is applied. This value is lower than the 1.0 K day$^{-1}$ used by Johnson et al. (2004) and Hill et al. (2008) because the
greater cooling rates result in an unstable cloud top height in our simulations which is undesirable as we require a consistent
cloud layer to isolate the cloud response due to the absorbing aerosol. A prescribed surface pressure of 1012.5 hPa is used, and
zonal and meridional geostrophic winds are 6.0 m s$^{-1}$ and -1.0 m s$^{-1}$, respectively. The radiation scheme is set up for consistency
with the FIRE campaign with a time–varying solar zenith angle for mid–July at the co–ordinates 33°N, 123°W. Radiation
calculations are performed every 30 seconds. Surface roughness is fixed at $2 \times 10^{-4}$ m and SST at 287.2 K.

**Table 1. Initial profiles used in the control simulations**

| Altitude (m) | Liquid–water potential temperature (K) | Total water mixing ratio (g kg$^{-1}$) |
|:---:|:---:|:---:|
| 0 | 287.5 | 9.0 |
| 600 | 287.5 | 9.0 |
| 601 | 297.0 | 5.5 |
| 750 | 300.0 | 5.5 |
| 1000 | 301.7 | 5.5 |
| 1500 | 303.2 | 5.5 |
| 2600 | 304.0 | 5.5 |




## 2.3 Setup of elevated–aerosol experiments

To simulate the effect of an elevated absorbing–aerosol layer above the cloud top, a layer of dry aerosol is prescribed, consisting of soot–like and water–soluble–like aerosol, representing predominantly absorbing and scattering species, respectively. The interaction of longwave and shortwave radiation with the aerosol layer results in localised heating rates that are coupled to the LEM. The prescribed aerosol layer properties include the height of layer base above the inversion layer (referred to as the cloud–aerosol gap), geometric thickness, mean single–scattering albedo (SSA), and AOD. These properties are set at the beginning of the experiment and applied during each call to the radiation scheme. Using the prescribed geometric thickness of the aerosol layer, a balance between the mass–mixing–ratio of soot and water–soluble aerosol is used to achieve the desired SSA and AOD throughout the simulation (see Appendix for more details on the method employed). In these experiments SSA is 0.9, which is towards the higher end of the range of SSA for biomass burning aerosol (Peers et al., 2016) and thus represents a relatively conservative value for the absorption of the aerosol layer.

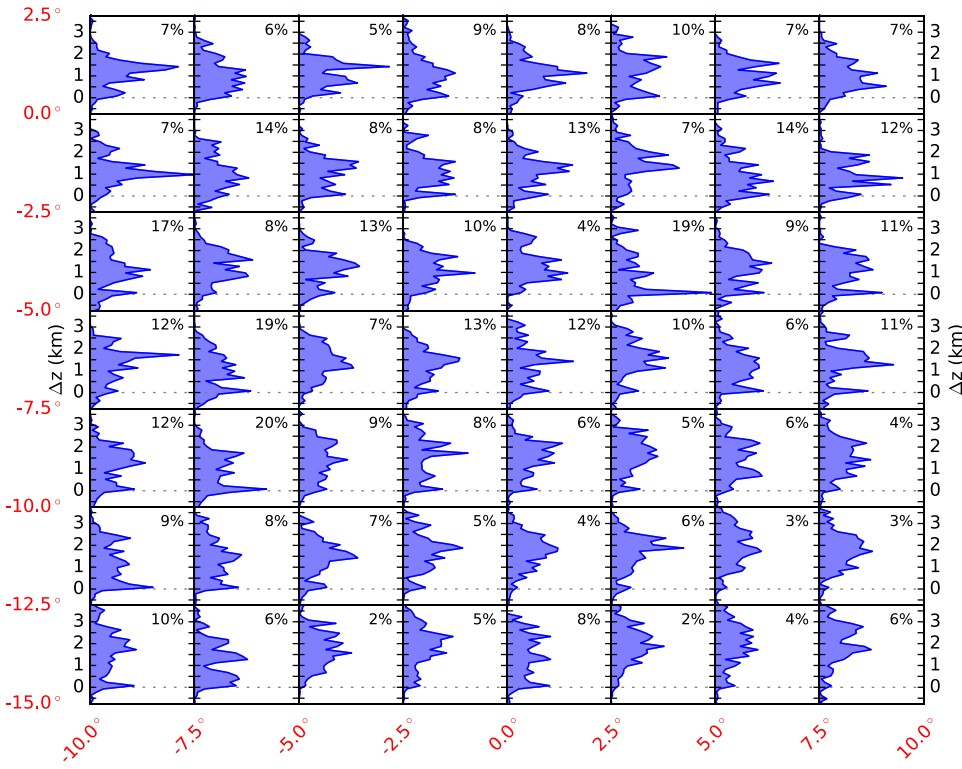

**Figure 1. Normalised frequency of occurrence of gap distance between cloud layer top and aerosol base heights from CALIOP for single layer coincidences of aerosol and cloud in the months of July, August, and September 2007–2016 over the southeast Atlantic (15°S to 2.5°N, 10°W to 10°E). Layer heights are binned from -1.5 to 5.5 km in 150 m increments and data in each grid has been normalised to the maximum frequency across the whole study area. The percentage of scenes where the aerosol layer base is below the cloud top height is shown in the top right of each subplot.**




Realistic cloud–aerosol gaps are needed for the elevated–aerosol experiments. They are taken from observations from the
CALIPSO Cloud–Aerosol Lidar with Orthogonal Polarization (CALIOP) instrument (5-km resolution, 532 nm Aerosol Layer
Product and Cloud Layer Product, v4.10, level 2 data) over the Southeast Atlantic Ocean (15°S to 2.5°N, 10°W to 10°E). The
distance $\Delta z$ between the retrieved cloud top and the aerosol base heights is determined from scenes where vertical profiles only
include a single layer of low cloud and a single layer of aerosol. Figure 1 shows the normalised frequency of occurrence of $\Delta z$
in 2.5-degree grids for all scenes within July, August, and September between 2007 and 2016. For the majority of scenes, the
layer of aerosol tends to be above, or directly above, the cloud top layer. There is considerable variation in $\Delta z$ at all locations;
to the north of the study area the peak $\Delta z$ is ~ 1 km, whereas to the south of the study area the peak $\Delta z$ is closer to 2 km. In
many regions there is a high frequency of the aerosol layer being in close proximity to the cloud top. Rajapakshe et al. (2017)
have shown that in the southeast Atlantic the CALIOP product overestimates the aerosol layer base height and layers are likely
much closer to the cloud layer than previously thought; they find that in 60% of their above–cloud–aerosol cases the absorbing
layer is less than 360 m above the cloud deck, therefore our $\Delta z$ is likely overestimated.

The CALIOP analysis (Fig. 1) suggests that elevated aerosol layers predominantly exist within 1500 m of the cloud top with
a common occurrence of layers in close proximity (< 150 m) to the cloud, and the study by Rajapakshe et al. (2017) suggests
the aerosol layers are predominantly within 360 m of the cloud top. In-line with this we focus on layers of absorbing layers
that range from directly above the cloud layer ($\Delta z = 0$ m) to elevated layers at $\Delta z = 500$ m, and we additionally examine the
role of the aerosol layer depth which, for a given AOD, will impact the vertical distribution and strength of the localised heat
perturbation.

A schematic of the experiments designed to investigate the sensitivity of the SDE and cloud diurnal cycle to key layer
properties, namely the AOD, geometric thickness, and the cloud–aerosol gap, is shown in Fig. 2. The first set investigates the
sensitivity of the SDE to the strength of the aerosol layer absorption. Following AOD observations by Chand et al. (2009), the
AOD of the layer is varied from 0.1 to 0.5 while keeping the geometric thickness constant at 200 m and the cloud–aerosol gap
at 50 m. The second set of experiments investigates the sensitivity of the cloud response to the geometric thickness of the
aerosol layer at constant AOD. This type of experiment is analogous to a satellite retrieval that estimates the AOD and aerosol
layer top but does not detect the lower extent of the aerosol layer. This is a known deficiency with retrievals made using
wavelengths that are strongly attenuated by biomass burning aerosol such as the 532 nm channel currently used in the
CALIPSO aerosol products (Rajapakshe et al., 2017). For these experiments the geometric thickness of the aerosol layer is
increased from 50 to 500 m with no cloud–aerosol gap and are effectively experiments with variable density of aerosol
particles, since with a fixed AOD the aerosol layer mass–mixing ratio decreases with increasing geometric thickness of the
layer. The final set of experiments investigates the impact of the cloud–aerosol gap by placing the aerosol layer from 0 to
500 m above the inversion layer while keeping geometric thickness and AOD constant. A full list of experiments performed





is presented in Table 2. We use one of the experiments, referred to as the base experiment, to provide an initial in-depth analysis
of the cloud and radiative response. In the base experiment (hatched experiment in Fig. 2) a 250 m thick absorbing aerosol
layer with an AOD of 0.2 is placed directly above the inversion layer.

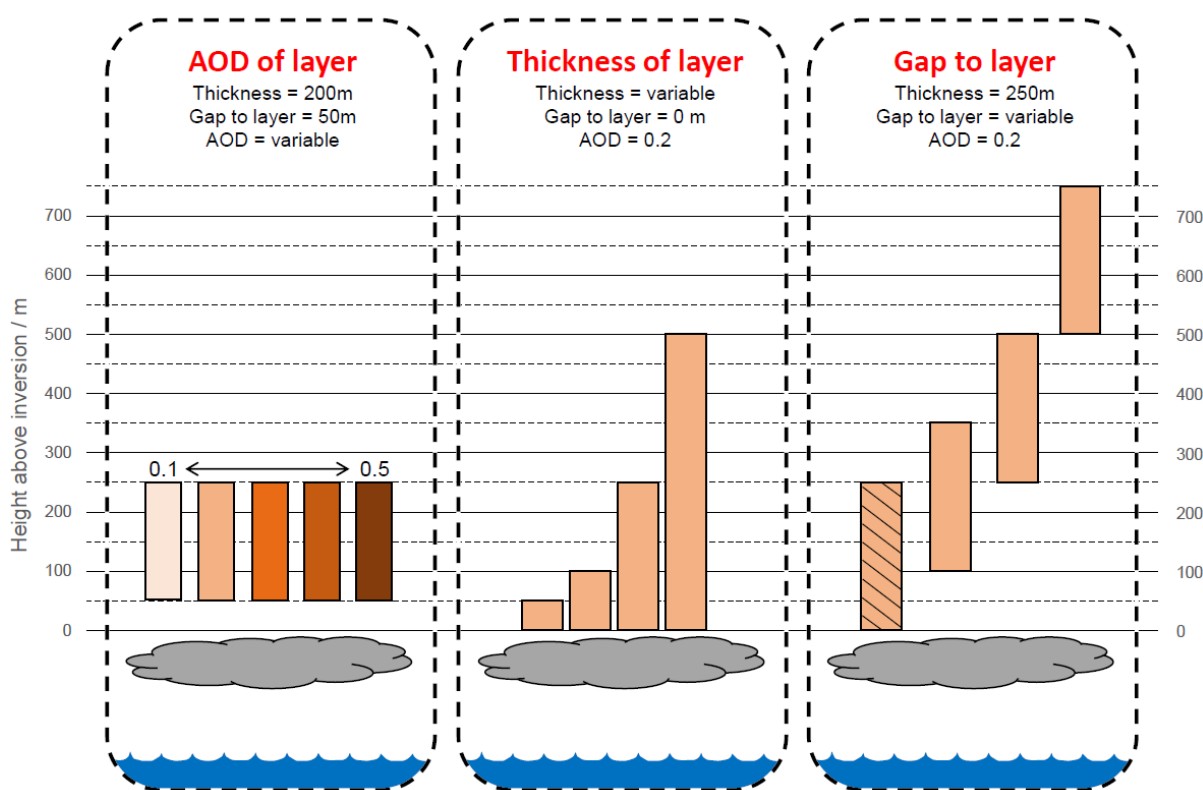

**Figure 2. Schematic showing the experiments performed for the aerosol–sensitivity simulations. The hatched experiment is named**
**the base experiment and is used to provide initial analysis of the semi–direct effect in Sect. 3.2. AOD stands for aerosol optical depth**
**and is given at a mid–band wavelength of 505 nm.**

The SDE is calculated as a residual of the difference in top–of–atmosphere net radiation ($F_{\text{TOA}}$) between the aerosol and no–
aerosol simulations, minus the direct radiative effect (DRE):

$$\text{SDE} = \left( F_{\text{TOA,no–aerosol}} - F_{\text{TOA,aerosol}} \right) - \text{DRE} \qquad (1)$$


where $F_{\text{TOA}}$ is calculated using the upward (↑) and downward (↓) fluxes of longwave (LW) and shortwave (SW) radiation:

$$F_{\text{TOA}} = F^{\downarrow}_{\text{TOA,SW}} - \left( F^{\uparrow}_{\text{TOA,SW}} + F^{\uparrow}_{\text{TOA,LW}} \right) \qquad (2)$$






DRE is calculated as the difference between $F_{TOA}$ and that obtained in a second, diagnostic, call to the radiation scheme with
the same profiles of liquid water, water vapour, and atmospheric gases, but without aerosol.

**Table 2. Breakdown of all experiments performed. AOD stands for aerosol optical depth and is given at a mid–band wavelength of**
**505 nm.**

| Type of experiment | Layer properties | | |
|---|---|---|---|
| | Cloud– aerosol gap (m) | Layer thickness (m) | Layer AOD |
| Variable AOD | 50 | 200 | **0.1** |
| | 50 | 200 | **0.2** |
| | 50 | 200 | **0.3** |
| | 50 | 200 | **0.4** |
| | 50 | 200 | **0.5** |
| Variable thickness | 0 | **50** | 0.2 |
| | 0 | **100** | 0.2 |
| | 0 | **250** | 0.2 |
| | 0 | **500** | 0.2 |
| Variable gap | **0*** | 250 | 0.2 |
| | **100** | 250 | 0.2 |
| | **250** | 250 | 0.2 |
| | **500** | 250 | 0.2 |

\* Base experiment used for initial analysis

## 3    Results

### 3.1    No–aerosol experiment

The no–aerosol experiment is initialised then run for fifteen days without the presence of an aerosol layer. The first five days
are used as a spin–up period that allows the BL to reach a steady state; the following three days (days 6, 7, and 8 of the
simulation) are shown in Fig. 3.

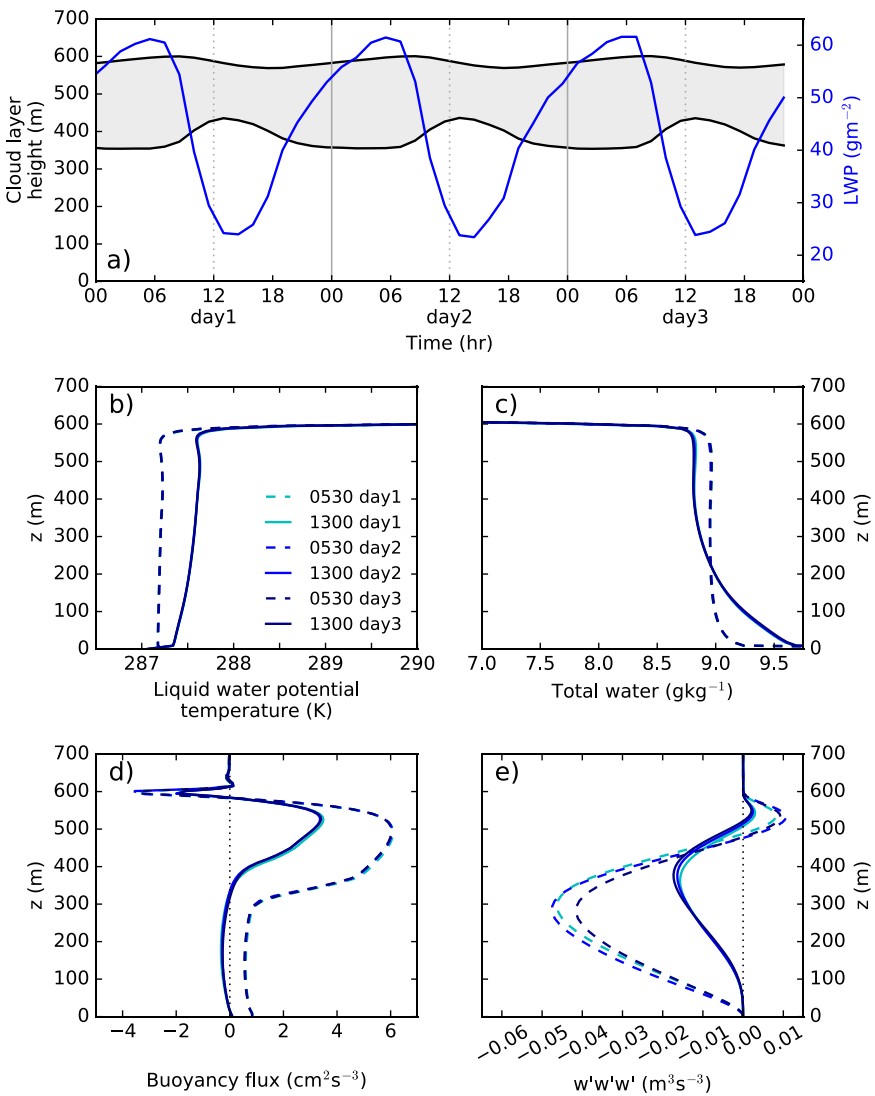


**Figure 3. Evolution of domain averaged cloud properties in the no–aerosol simulation including: a) cloud top and base (black lines; left axis), and liquid water path (blue line; right axis); and vertical profiles taken at 0530 (dashed lines) and 1300 (solid lines) on each day for b) liquid water potential temperature, c) total water mass mixing ratio, d) buoyancy flux, and e) the perturbation in mean vertical velocity w'w'w'.**

The no–aerosol experiment produces a cloud-topped BL with strong diurnal variations. During the daytime, cloud top height decreases and cloud base height increases, thinning the cloud and producing a diurnal cycle of LWP that reaches a maximum of 60 g m$^{-2}$ at dawn and a minimum of 25 g m$^{-2}$ just after midday (Fig. 3a). The diurnal cycle can be separated into a growth phase between 1400 and 0600, and a decay phase between 0700 and 1300. The growth phase is driven by pronounced buoyancy production during the night (Fig. 3d) from longwave cloud–top cooling and evaporative cooling of entrained air, which drives strong eddies below cloud (Fig. 3e). During the daytime, solar heating reduces cloud–top negative buoyancy through an offset





in the longwave cooling and reduces surface–driven positive buoyancy through weakened surface–to–atmosphere gradients.
This weakens the BL circulation and prevents mixing throughout the BL and promotes a decoupled state in which the flux of
moisture from the surface to the cloud is insufficient to maintain the cloud base height, as evident from the non–constant BL
profiles of $\theta_l$ (Fig. 3b) and $q_t$ (Fig. 3c) at 1300 hours. The weakened flux and solar heating of the cloud drives the lifting
condensation level upwards and causes the cloud base to increase with height, producing the decay phase. During the daytime
weakened BL eddies are unable to 'push' against the subsidence at the BL top, which decreases the BL depth and cloud top
height. Due to the different processes that control the cloud top and cloud base diurnal variations, the cloud top height minimum
occurs about 2 hours after the cloud base reaches its maximum. The cloud layer, LWP and thermodynamic profiles in Fig. 3
(a – e) show very little change over the three days of the simulation and present a stratocumulus deck with a consistent diurnal
cycle in a steady state. This provides a suitable simulation to use as control for the elevated–aerosol experiments.

## 3.2     Cloud response to elevated aerosol layer in the base experiment

We begin with the base experiment (hatched experiment in Fig. 2) where a 250 m thick absorbing aerosol layer with an AOD
of 0.2 is placed directly above the inversion layer. Following a five-day spin–up period without aerosol, the simulation runs
for a further ten days with the aerosol layer present. The domain–averaged cloud response following the introduction of aerosol
is shown in Fig. 4 and compared to the no–aerosol simulation.
The simulations show that the absorbing aerosol drives changes in the diurnal cycle of cloud depth and LWP, predominantly
through changes in the cloud base height. The presence of the absorbing aerosol drives a decrease in cloud top height (Fig. 4a)
which occurs predominantly in the afternoon and evening and is indicative of a decrease in entrainment across the inversion
layer. During the initial two days the cloud base (Fig. 4a) decreases in altitude ~10 m more than the cloud top resulting in a
thicker cloud, however from day three onwards there is less growth of the cloud throughout the evening and early morning,
followed by less thinning throughout the day. Compared to the cloud in the presence of no aerosol, the introduction of the
absorbing aerosol layer results in relatively less LWP (Fig. 4b) during the growth phase of the cloud and more LWP during
the decay phase.
The SDE (Fig. 4d) has a strong diurnal cycle that is directly driven by modifications to the cloud albedo diurnal cycle (Fig.
4c) and shows considerable sensitivity to the LWP response during the cloud decay phase around midday. In the first three
days the albedo response is positive from mid–morning to the late afternoon. This drives an overall negative daily mean SDE.
The length of time with a positive albedo response gets shorter as the simulation progresses, driving an increasingly positive
SDE in the morning that cancels out, on a daily mean, the negative SDE in the afternoon. Consequently, the daily mean SDE
is negative for the initial three days but almost net zero SDE from the fourth day onwards.



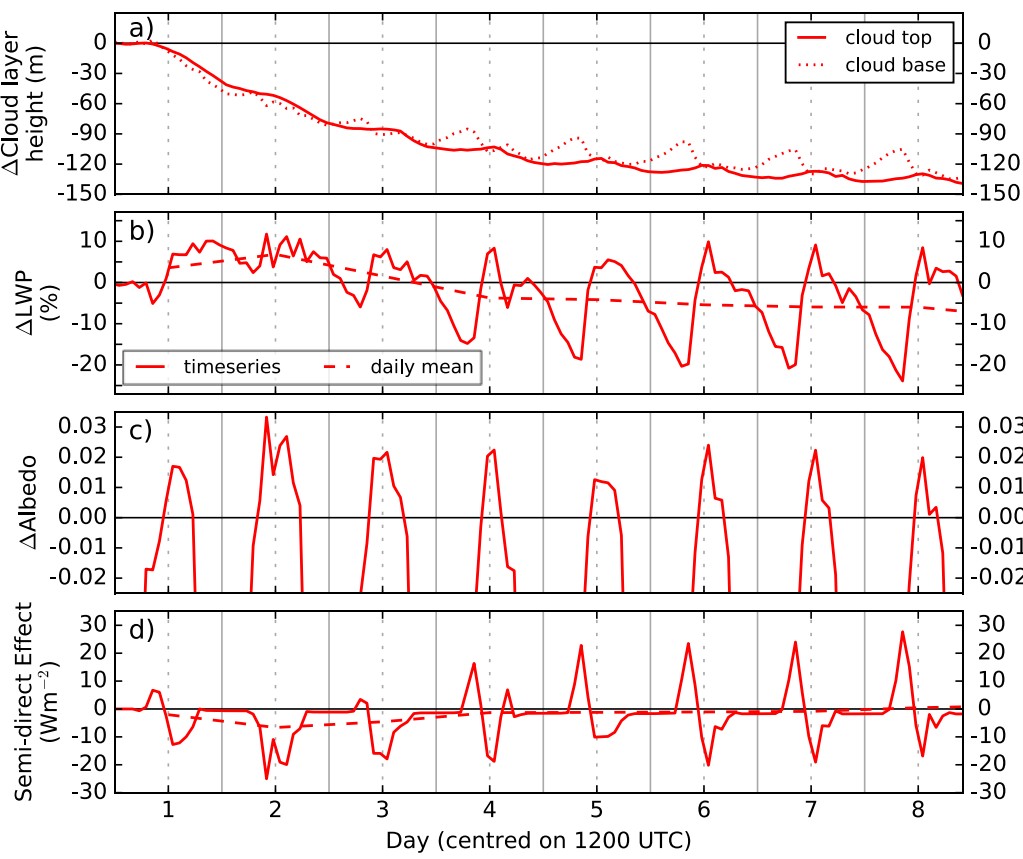


**Figure 4. 10–day timeseries of domain–averaged cloud response to a layer of aerosol directly above the boundary layer inversion**
**with an aerosol optical depth of 0.2 and geometric thickness of 250 m. Plots show the repsonse from a) cloud top height (solid line)**
**and cloud base height (dotted line), b) cloud liquid water path (LWP), c) albedo, and d) the semi–direct effect. Solid lines in b), c),**
**and d) show the timeseries of the response and dashed lines in b) and d) show the daily mean.**


The cloud response and SDE are therefore markedly different in the initial phase compared to the steady–state that is reached
after 6 or 7 days following the introduction of the absorbing aerosol layer. In that steady–state phase the BL depth has decreased
by ~130 m (~20%) and the diurnal cycle response in cloud thickness has stabilised. This suggests there are timescales in the
response to the introduction of the aerosol layer: a short–term response that can be interpreted as a rapid adjustment of the
humidity profile, and longer–term response that can be interpreted as a new equilibrium state for the BL sources of moisture,
turbulence, and heat.





This study focuses on the initial response because it is more relevant for real–world understanding as the aerosol perturbation
is unlikely to remain constant for several days, and the lifetime of stratocumulus decks is generally on the order of a few days
only. However, the steady–state response provides insight into the key drivers behind the BL modifications.
**3.2.1    Initial response in the base experiment**
The domain–averaged timeseries of the response in the first three days following the introduction of the aerosol layer (days 6,
7, and 8 of the simulation) are shown in Fig. 5. The initial response of the cloud to the elevated aerosol layer is driven by the
weakening of the entrainment rate ($w_e = dz_{\mathrm{cloudtop}} \ / \ dt - w_s$) and subsequent increase in the RH below cloud which acts to
produce a thicker cloud in the first two days. Solar radiation heats the elevated absorbing aerosol layer above the inversion
layer. Strengthening of the temperature inversion at the top of the BL drives a weakened $w_e$ (Fig. 5b) which causes the BL
depth to decrease (Fig. 5a). Simultaneously, there is an increase in RH below cloud (Fig. 5d), which allows the cloud base
height to decrease (Fig. 5a) and the LWP to increase (Fig. 5c); this response continues for the first two days, after which the
LWP starts to display a diurnal response with a decrease in LWP during the night and an increase in the afternoon. The increase
in RH occurs due to the weakened $w_e$ which reduces the amount of warm dry FT air that is mixed into the BL and allows the
cloud layer to maintain a higher RH.

The thinner cloud on the morning of the third day is driven by changes to the supply of moisture to the cloud layer. The
enhanced RH below cloud (caused by an increase in water vapour) and weakened vertical motions (Fig. 5g) drive a strong
reduction in surface evaporation as demonstrated by the decrease in latent heat flux (LHF; Fig. 5e), especially during the night.
By the end of day three the LHF at the surface has reduced by 20% and the total column water content (Fig. 5f) has reduced
by 10%. During the night when the BL is well mixed this reduction in total water content prevents the cloud from developing
to the same extent as in the no–aerosol simulation, resulting in a thinner cloud when the sun rises. This process is amplified by
the reduced BL dynamics which will weaken the flux of moisture from the sub–cloud region to the cloud.

The thicker cloud on the afternoon of the third day is driven by relatively stronger coupling with the surface moisture fluxes
at midday, which produces a slightly thicker cloud and a negative SDE (Fig. 5h). Under no-aerosol conditions, shortwave
absorption by the cloud stabilises the cloud layer during the day, which results in a degree of decoupling between the surface
layer and cloud base (Fig. 3). When an elevated absorbing aerosol layer is present, the decrease in cloud layer height, following
the BL depth decrease, allows better coupling to the surface, which becomes increasingly important around midday when BL
dynamics are weakest (Fig. 5g). The enhanced source of moisture to cloud base, along with weakened entrainment of dry FT
air, prevents the cloud from thinning to the same extent. Although the change in LWP is only 2–3 g m$^{-2}$, this amounts to a 10%
increase, which helps drive a strong negative SDE at midday and early afternoon.




**Figure 5. 3–day timeseries showing the initial domain averaged cloud response to a layer of absorbing aerosol in the base**
**experiment In the first column the black dashed lines refer to the control experiment (no–aerosol) and solid blue lines to the**
**experiments with the aerosol layer present. The second column shows the cloud response (red solid line). The plots show a) the**
**altitude of the cloud base and top, b) the entrainment rate $w_e$, c) the liquid water path (LWP), d) the mean relative humidity (RH)**
**below cloud base, e) the surface latent heat flux (LHF), f) the total mass of water in the boundary layer (BL) column, g) the mean**
**squared BL vertical velocity perturbation ($w'w'$), and h) the semi–direct effect.**




The analysis of the initial cloud response shows that the first two days are characterised by a general thickening of the cloud
driven by the reduction in $w_e$ across the temperature inversion and subsequent enhanced RH profile below cloud. The weakened
$w_e$, BL dynamics, and moisture flux from the surface begin to dry the BL resulting in less cloud growth overnight, whilst the
lower cloud base enhances coupling to the surface moisture fluxes during the middle of the day, and less cloud decay.

**3.2.2    Steady–state response in the base experiment**
The final three days of the 15–day base experiment provide a mean diurnal cycle of the cloud response. The steady–state
response of the cloud to the elevated aerosol layer, shown in Fig. 6, shows strong similarities to the third day of the initial
response: the growth phase of the cloud (Fig. 6b) is weakened, producing a thinner cloud in the morning, and the decay phase
of the cloud (Fig. 6b) is weakened, producing a thicker cloud in the early afternoon. This modification to the diurnal cycle of
the cloud is driven by an increased coupling between surface moisture flux and cloud base during the daytime and an overall
decrease in total water content and weakened dynamics overnight. The decrease in cloud layer height allows better mixing
beneath the cloud base, which enhances the evaporation of moisture from the surface between 0900 and 1500 (Fig. 6d); this is
evident from the lack of a diurnal cycle in below–cloud RH (Fig. 6c), which usually occurs due to poor mixing, and the
strengthened BL dynamics at midday (Fig. 6f).

The weakened cloud growth phase overnight occurs due to a 15% reduction in total water content of the BL (Fig. 6e) and a
reduction in mean BL vertical motions overnight of ~20%, indicated by the mean perturbation to vertical velocity in the BL
($w'w'$) in Fig. 6f. The reduction in $w_e$ (Fig. 6a) and subsequent changes to below–cloud water vapour set up a positive feedback
mechanism with BL dynamics: vertical motions in the BL are considerably weakened throughout the night and slightly
strengthened at midday. Although there is a decrease in LWP there is no systematic impact to the cloud–top longwave cooling
due to its weak sensitivity to LWP above 50 g m$^{-2}$ (van der Dussen et al., 2013; Garrett and Zhao, 2006). The weakened BL
circulation is therefore due to a reduction in evaporation and associated cooling of entrained air, which weakens cloud–top
buoyancy production. These combined changes result in reduced vertical motions within the BL, which reduce surface
evaporation, cloud LWP, and buoyancy production from condensation at cloud base, which allow the reduced vertical motions
to persist. A partial offset to this process occurs during midday when stronger coupling to the surface results in enhanced
transport of water vapour to the cloud base.

The steady–state response establishes itself by the third day of the simulation. The daily mean steady–state SDE (Fig. 6g)
results from a balance between the degree to which the BL total column water content has decreased, producing a positive
SDE in the morning, and the degree to which the midday coupling is enhanced, producing a negative SDE in the afternoon. In
both cases modifications to BL depth, and thus $w_e$, play a significant role in cloud response and SDE.




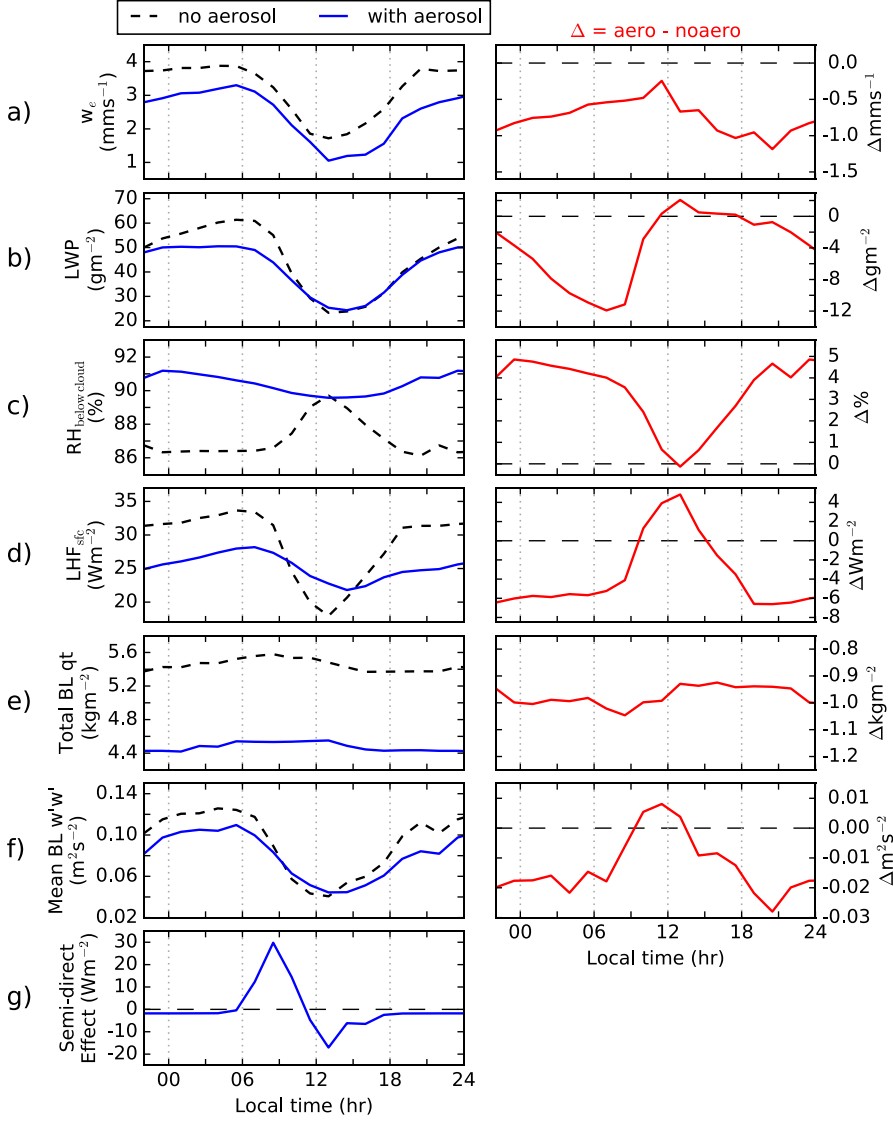


**Figure 6. Domain averaged cloud response to a layer of absorbing aerosol directly above the inversion in the base experiment (0 m cloud–aerosol gap, 250-m thick layer, and AOD of 0.2) for the mean diurnal cycle using the final three days of the 15–day simulation. In the first column the black dashed lines refer to the control experiment (no–aerosol) and solid blue lines to the experiments with the aerosol layer present. The second column shows the cloud response (red solid line). The plots show a) the entrainment rate $w_e$, b) the liquid water path (LWP), c) the mean relative humidity (RH) below cloud base, d) the latent heat flux (LHF) from the surface, e) the total mass of water (vapour + liquid) in the boundary layer (BL) column, f) the mean squared BL vertical velocity perturbation ($w'w'$), and g) the semi–direct effect.**

365





### 3.3 Sensitivity of initial response to aerosol layer properties

Figure 7 shows timeseries for the aerosol layer–sensitivity experiments. In this analysis the inversion strength $\Delta\theta_l$ is determined between altitudes $z_{upper}$ and $z_{lower}$ defined as:

$$\begin{cases} z_{lower} = z \text{ at } 0.025 \cdot z_{max} \text{ below } z_{max} \\ z_{upper} = z \text{ at } 0.25 \cdot z_{max} \text{ above } z_{max} \end{cases} \tag{3}$$

where $z_{max}$ is the altitude at which the maximum gradient in $\theta_l$ occurs. The upper value is determined at a lower threshold to limit spurious values occurring from aerosol layers close to the inversion layer that impact $\theta_l$.

### 3.3.1 Cloud response

The majority of experiments show a positive spike in SDE (Fig. 7d, i and n) just after midday on the first day. This occurs due to the lag–time in response between the impact on cloud top height, which is driven by $w_e$, and the cloud base, which is driven by changes in sub–cloud RH. This lag occurs due to weaker coupling of the cloud and sub–cloud layers and therefore poorer BL mixing around midday (see Fig. 3). This small decrease in LWP and subsequent positive SDE is consistent throughout the range of experiments but mitigated in some cases due to rapid impacts on the LWP (i.e., the 50 m thickness experiment) that occur before BL coupling weakens. This result suggests the specific timing of the incoming aerosol plume may play a role in the cloud response and SDE on the first day.

Geometrically thinner aerosol layers equate, for a given AOD, to a greater aerosol mass mixing ratio and therefore stronger heating. This results in a stronger inversion layer (Fig. 7a) and stronger modification to the LWP response (Fig. 7c) and SDE (Fig. 7d), especially on the first day. This produces a stronger inversion layer, weaker $w_e$, and a decrease in BL depth (Fig. 7b). For the two thinnest layers the cloud top height decreases at a faster rate during the day than at night, which correlates with the peak heat perturbation. For thicker layers the heat perturbation extends further into the night; this corresponds with the delay in time for the heating towards the top of the layer to reach the inversion layer and drives a steadier reduction in BL depth when compared to the thinner layers. By the third day the BL has started to adjust and less dependence on aerosol layer thickness is apparent, however the thinner layers cause the BL to dry out at a quicker rate, thus producing a stronger positive SDE on the morning of the third day.

392



**Figure 7. 3–day timeseries showing the sensitivity of the initial cloud response to the properties of the elevated absorbing aerosol layer. The three columns correspond to experiments where systematic changes have been made to the aerosol layer thickness (a – e), cloud–aerosol gap (f – j), and aerosol layer AOD (k – o).**


Increasing the cloud–aerosol gap leads to a weaker and increasingly delayed maximum cloud top height (Fig. 7g) and LWP

response (Fig. 7h) driven by changes in peak strengthening of the inversion (Fig. 7f); this is most pronounced in the first two

days. Only aerosol layers directly above the inversion trigger a considerable cloud response on the first day because of the




relatively rapid strengthening of the inversion layer and weakening of $w_e$ which forces the cloud top downwards more rapidly
than the RH profile can adjust, resulting in a deeper cloud base. On the second day a cloud response is seen with gaps up to
100 m and by the third day all gaps lead to a response in cloud LWP. The delay in response is driven by the delay in the
inversion layer strengthening. In the free troposphere the advection of the heat perturbation is driven by subsidence, therefore,
greater cloud–aerosol gaps require more time for the heat perturbation to reach the cloud top. Simultaneously longwave cooling
acts to weaken the heat perturbation throughout its advection, which drives a relatively weaker strengthening of the temperature
inversion as the cloud–aerosol gap increases.

The initial cloud top response (Fig. 7l) displays a strong dependence on the AOD of the aerosol layer throughout the three days
with greater AOD resulting in a greater response. As with geometric layer thickness, larger AODs absorb more radiation and
drive a stronger heat perturbation and inversion strength (Fig. 7k). So larger AODs result in a thicker cloud and a more negative
SDE. On the third day layers with the largest AODs, which have had the greatest impact on cloud top height and $w_e$, exhibit a
considerably thinner cloud, driving an increasingly positive SDE in the morning.

In summary, the layer–sensitivity experiments show that on the first day the initial response is for the cloud top to drop quicker
than the cloud base, resulting in a thinner cloud and a positive SDE in the morning, the magnitude of which is primarily driven
by the proximity of the aerosol layer with the cloud top. With no gap between the inversion at cloud top and aerosol layer, the
afternoon of the first day is characterised by a thicker cloud and negative SDE which increases in magnitude for stronger heat
perturbations. The second day is generally characterised by an increase in the LWP at midday which drives a negative SDE
and is dependent on the location and properties of the aerosol layer. By the third day a consistent pattern occurs: the cloud is
consistently thinner in the morning and thicker at midday, the magnitude of which is dependent on the strength of the
perturbation.
**3.3.2    Radiative response**
Figure 8 shows timeseries of the daily mean radiative effects for the layer–sensitivity experiments. The immediate radiative
response following the introduction of the absorbing aerosol layer is primarily dependent on the distance between the inversion
layer and aerosol layer base. When there is no cloud–aerosol gap the increase in LWP results in a negative SDE; thinner layers
and larger AODs increase the inversion layer strengthening and drive a stronger negative SDE on the first day. When any
cloud–aerosol gap is present there is little LWP response on the first day due to the delayed inversion layer strengthening,
however, all experiments with a gap present are characterised by a small positive SDE. For the experiments with a 50-m gap
(variable AOD experiments) the delay is short enough that there is an increase in LWP in the evening of the first day (Fig. 8i).

On the second and third day the SDE is negative for all experiments; the magnitude of the SDE increases for thinner layers,
closer to the inversion layer. When a cloud–aerosol gap is present the AOD tends to have little impact on the magnitude of the





SDE. The rate at which the BL moisture content decreases, itself a factor of how strongly $w_e$ is perturbed, results in variations
in which day the peak SDE occurs. In experiments with gaps smaller than 100 m the maximum SDE is reached on the second
day, whereas for gaps larger than or equal to 100 m the maximum occurs on the third day. In all experiments the third day is
characterised by a decrease in the daily mean LWP response which is primarily driven by less cloud growth overnight and in
the morning (see Fig. 7c, h and m) and becomes more pronounced as the temperature inversion strengthens. The thinner cloud
in the morning helps to shift the daily mean SDE towards zero.

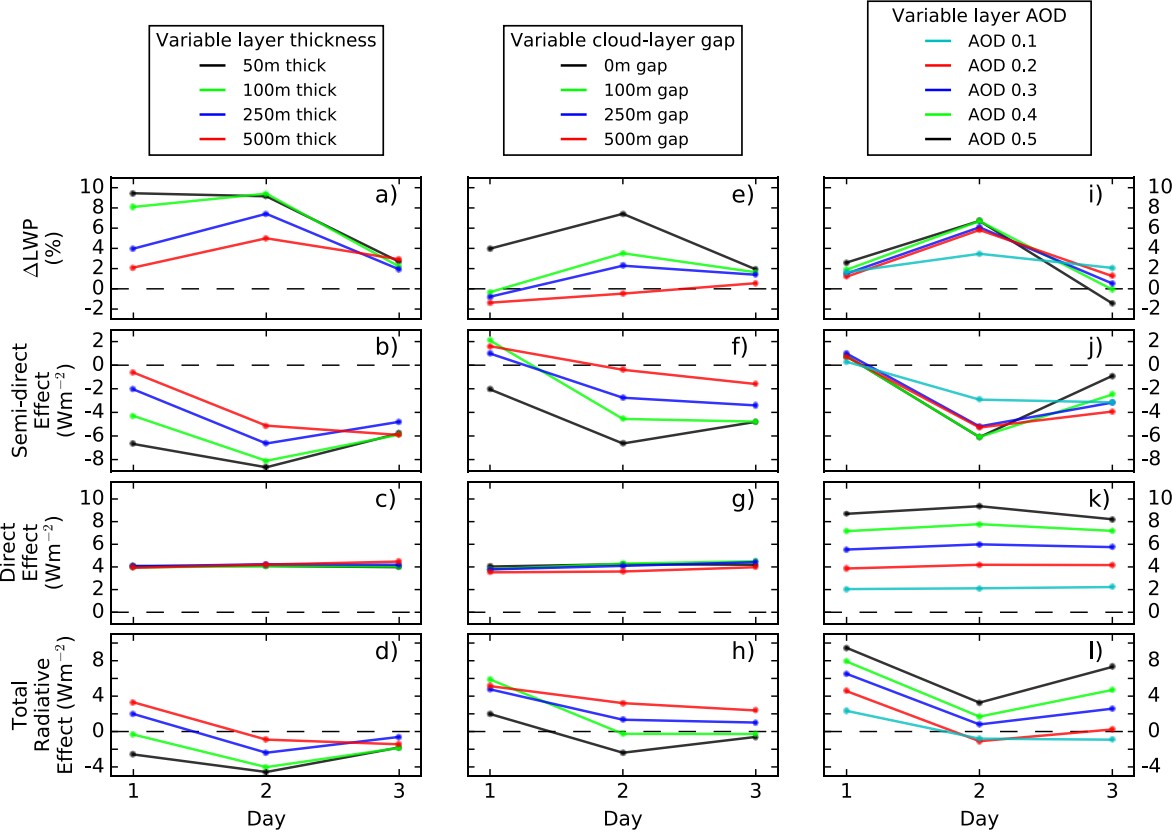


**Figure 8. Daily mean radiative impact to the elevated aerosol layer properties over the initial three days following the introduction**
**of the aerosol layer for systematic changes to a) – d) aerosol layer thickness, e) – h) cloud–aerosol gap, and i) – l) aerosol optical**
**depth of layer.**

The properties of the aerosol layer have a considerable impact on the total radiative effect, calculated as the sum of the DRE
and SDE (Fig. 8d, h, and l). Generally, the SDE acts to counteract the positive DRE and in some cases results in an overall
negative total radiative effect. In all experiments the total radiative effect is sensitive to the layer properties, whereas DRE is
only sensitive to the layer AOD. In many instances the SDE is greater in magnitude than the DRE, with the second day
constituting the period of time with the greatest impact. The relative insensitivity of the SDE to changes in AOD suggest that





layers with a moderate AOD (~ 0.2) may have the strongest overall radiative impact due to the relatively low DRE; however,
the behaviour may change for increasing gaps.

The results of the experiments are summarised in Table 3 with the daily mean SDE alongside the means for the periods before
and after midday. The daily mean SDE is only consistently negative throughout the three days when there is no cloud–aerosol
gap. This result is consistent with Johnson et al. (2004) who similarly found a negative SDE for a ~1000 m layer of absorbing
aerosol (AOD of 0.2, SSA of 0.88) directly above the inversion layer. Johnson et al. (2004) calculated a mean SDE of -9.5 Wm$^{-2}$
and a mean DRE of 10 Wm$^{-2}$. These magnitudes are greater than in this study but similarly show the SDE is of approximately
equal magnitude to the DRE and of opposite signs. Our results also show that geometrically thin, but optically thick, aerosol
layers will have a stronger forcing than a thicker layer with the same AOD due to a stronger localised heat perturbation; this
effect is most prominent on the first day. When a gap to the aerosol layer base is present our results show that the short–term
SDE is likely to be positive but then becomes negative once the BL has been mixed, which usually occurs during the first night
when BL mixing occurs, highlighting a sensitivity to the specific arrival timing of the incoming plume. On the second and
third day the magnitude of the SDE then depends on the AOD, cloud–aerosol gap, and aerosol layer thickness.

**Table 3. Mean semi–direct effect (in Wm$^{-2}$) for each of the aerosol experiments shown in Fig. 2 and Table 2. Mean values are**
**presented for each day (Daily), between 0000 and 1200 hours (am), and between 1200 and 2400 (pm). For the daily mean,**
**increasingly negative values are shaded in blue and increasingly positive in red.**

| Type of experiment | gap | dz | AOD | Day 1 Daily | am | pm | Day 2 Daily | am | pm | Day 3 Daily | am | pm |
|---|---|---|---|---|---|---|---|---|---|---|---|---|
| Variable gap | **0** | 250 | 0.2 | -2 | 1 | -5 | -7 | -6 | -7 | -5 | -4 | -6 |
| | **100** | 250 | 0.2 | 2 | 4 | 0 | -5 | -5 | -4 | -5 | -5 | -5 |
| | **250** | 250 | 0.2 | 1 | 2 | 0 | -3 | -3 | -3 | -3 | -3 | -4 |
| | **500** | 250 | 0.2 | 2 | 1 | 2 | 0 | 1 | -1 | -2 | -3 | 0 |
| Variable thickness | 0 | **50** | 0.2 | -7 | -2 | -12 | -9 | -5 | -13 | -6 | -2 | -10 |
| | 0 | **100** | 0.2 | -4 | -1 | -8 | -8 | -5 | -11 | -6 | -2 | -10 |
| | 0 | **250** | 0.2 | -2 | 1 | -5 | -7 | -6 | -7 | -5 | -4 | -6 |
| | 0 | **500** | 0.2 | -1 | 2 | -3 | -5 | -4 | -6 | -6 | -5 | -7 |
| Variable AOD | 50 | 200 | **0.1** | 0 | 2 | -1 | -3 | -3 | -3 | -3 | -2 | -4 |
| | 50 | 200 | **0.2** | 1 | 2 | 0 | -5 | -5 | -6 | -4 | -4 | -4 |
| | 50 | 200 | **0.3** | 1 | 2 | 0 | -5 | -5 | -5 | -3 | -1 | -5 |
| | 50 | 200 | **0.4** | 1 | 2 | -1 | -6 | -4 | -8 | -2 | 0 | -5 |
| | 50 | 200 | **0.5** | 1 | 3 | -1 | -6 | -4 | -8 | -1 | 3 | -5 |




Table 3 highlights the diurnal variations in the SDE. The SDE is generally more negative after midday but that contrast varies
with aerosol layer properties. Geometrically thin, optically thick layers, directly above the inversion layer display the strongest
contrast with the daily mean SDE dominated by the mean after midday. When a gap is present there is less contrast and both
time periods are generally representative of the daily mean, until the BL begins to dry out significantly in the high AOD
experiments. These results demonstrate that there are often strong diurnal variations in the SDE which are sensitive to the
aerosol layer properties and suggest that observations of the SDE made within a small window of time, e.g., those from polar
orbiting satellites, may be unrepresentative of the daily mean SDE.
**3.4    Sensitivity to boundary layer and cloud properties**
This section investigates the robustness of the results and conclusions from Sect. 3.3. The parameter space considered in this
section includes previous LEM studies, such as Hill and Dobbie (2008) and Johnson et al. (2004), and the range of
environmental forcings observed within marine stratocumulus regions.

The first set of sensitivities focus on the model setup and includes no precipitation from the cloud (*noRain*) and an enhanced
large–scale advective heat tendency of -0.5 Kday$^{-1}$ (*05cool*).
• In the *noRain* setup the production of precipitation is switched off. Stratocumulus frequently produce precipitation in
the form of drizzle (Leon et al., 2009) yet studies often simplify simulations by focusing on non-precipitating
stratocumulus (e.g., Hill and Dobbie, 2008; Johnson et al., 2004). Precipitation redistributes moisture from the cloud
layer to the sub–cloud layer, promoting BL stability and acting to reduce BL dynamics and cloud LWP (Ackerman
et al., 2009).
• In the *05cool* sensitivity, the magnitude of the large–scale advective heat tendency is increased from -0.1
to -0.5 Kday$^{-1}$. That parameter accounts for the equatorward transport of the large–scale air mass and is negative in
subtropical marine regions. This value can be estimated using large-scale reanalyses (e.g., Johnson et al., 2004) or
used as a balancing term to prevent subsidence heating (e.g., Duynkerke et al., 2004) and represents a degree of
variability in LES setups.

The second set of sensitivities focuses on properties of the BL that may impact the diurnal cycle and maintenance of the cloud.
• In the *SST-1K* and *SST+1K* setups, SST is decreased and increased by 1K, respectively, while keeping the BL depth
at 600 m. Stratocumulus decks in the Atlantic and Pacific Oceans are observed over a wide range of sea surface
temperatures (Sandu and Stevens, 2011; Wood, 2012). As the SST increases the differential temperature across the
surface–air boundary increases, resulting in more pronounced surface moisture and sensible heat fluxes.
• The *wetFT* setup increases the mass mixing ratio of water vapour in the FT by +0.4 g kg$^{-1}$ to assess the impact of the
water vapour content of the entrained air on the SDE. Trajectory analyses from the Pacific and Atlantic Oceans by





Sandu et al. (2010) show that the mass mixing ratio of water vapour in the FT varies spatially and temporally, ranging
from 1.0 to 7.5 g kg$^{-1}$ at 700 hPa; this result is supported by in–situ data summarised by Albrecht et al. (1995).
•    The *800-m* and *1000-m* setups increase the height of the temperature inversion by 200 and 400 m, respectively, by
changing the large–scale divergence rate and initial profiles of $\theta_l$ and $q_t$, while keeping SST constant at 287.2 K.
Observations show that cloud top heights in regions of semi–permanent stratocumulus coverage (southeast Atlantic,
southeast Pacific, and northeast Pacific) typically range from ~500 to ~1500 m (Muhlbauer et al., 2014; Painemal et
al., 2014; Wyant et al., 2010) with variations driven by SST and subsidence.

To isolate the cloud response due to the aerosol layer, the cloud–sensitivity experiments are initialised using profiles that
produce an approximately constant stratocumulus cloud layer at the top of the BL following the method described in Sect. 2.2.
Table 4 shows the resulting initial profiles and large–scale divergence rates for each setup. The daily mean SDE on day 2
following the introduction of the absorbing aerosol layer (day 7 of the simulation) is shown in Table 5 for each setup and
aerosol experiment. For the *control* setup the SDE values are the same as shown in Fig. 8.

**Table 4. Initial profiles of liquid–water potential temperature ($\theta_l$ in K) and total liquid mass–mixing ratio ($q_t$ in g kg$^{-1}$) against**
**altitude ($z$ in m) for each cloud–sensitivity setup. Values in parentheses indicate the large–scale divergence rate ($D$ in s$^{-1}$) used for**
**each setup. All setups result in a stable stratocumulus cloud deck at the top of the boundary layer.**

| | noRain ($5.4 \times 10^{-6}$) | | 05cool ($6.2 \times 10^{-6}$) | | SST-1K ($4.75 \times 10^{-6}$) | | SST+1K ($5.75 \times 10^{-6}$) | | wetFT ($5.25 \times 10^{-6}$) | | 800m ($4.0 \times 10^{-6}$) | | | 1000m ($2.75 \times 10^{-6}$) | | |
|---|---|---|---|---|---|---|---|---|---|---|---|---|---|---|---|---|
| $z$ | $\theta_l$ | $q_t$ | $\theta_l$ | $q_t$ | $\theta_l$ | $q_t$ | $\theta_l$ | $q_t$ | $\theta_l$ | $q_t$ | $z$ | $\theta_l$ | $q_t$ | $z$ | $\theta_l$ | $q_t$ |
| 0 | 287.5 | 9.0 | 287.3 | 9.0 | 286.5 | 8.6 | 288.3 | 9.4 | 287.3 | 9.0 | 0 | 287.3 | 9.0 | 0 | 287.3 | 9.0 |
| 600 | 287.5 | 9.0 | 287.3 | 9.0 | 286.5 | 8.6 | 288.3 | 9.4 | 287.3 | 9.0 | 800 | 287.3 | 9.0 | 1000 | 287.3 | 9.0 |
| 601 | 297.0 | 5.5 | 296.0 | 5.5 | 296.0 | 5.5 | 297.2 | 5.5 | 297.0 | 5.9 | 801 | 297.0 | 5.9 | 1001 | 297.0 | 5.9 |
| 750 | 300.0 | 5.5 | 299.0 | 5.5 | 300.0 | 5.5 | 300.0 | 5.5 | 299.5 | 5.9 | 900 | 299.5 | 5.9 | 1100 | 299.5 | 5.9 |
| 1000 | 301.7 | 5.5 | 300.3 | 5.5 | 301.7 | 5.5 | 301.7 | 5.5 | 301.5 | 5.9 | 1200 | 301.5 | 5.9 | 1300 | 301.5 | 5.9 |
| 1500 | 303.2 | 5.5 | 301.5 | 5.5 | 303.2 | 5.5 | 303.2 | 5.5 | 302.6 | 5.9 | 1700 | 302.6 | 5.9 | 1900 | 302.6 | 5.9 |
| 2600 | 304.0 | 5.5 | 302.8 | 5.5 | 304.0 | 5.5 | 304.0 | 5.5 | 303.8 | 5.9 | 2600 | 303.8 | 5.9 | 2600 | 303.8 | 5.9 |


### 3.4.1    Sensitivity to model setup

The removal of precipitation results in stronger BL dynamics and a greater peak in LWP (+15 g m$^{-2}$). Compared to the *control*
setup the *noRain* setup is characterised by a consistent strengthening of the SDE at +1 Wm$^{-2}$ when a cloud–aerosol gap is
present and up to +3 Wm$^{-2}$ when there is no gap. In the *control* setup the presence of the aerosol layer increases cloud LWP,
which is partially offset by an increase in precipitation. In the *noRain* setup that partial offset is not allowed, resulting in
relatively enhanced LWP response and SDE.






Increasing the cooling rate of the large–scale advective heat tendency produces stronger buoyancy production and BL
dynamics, which are balanced by stronger subsidence ($D = 6.2 \times 10^{-6}\,\text{s}^{-1}$) in order to maintain a 600 m BL depth. An enhanced
cloud top entrainment of warm dry air is balanced by enhanced flux of vapour from below–cloud and surface LHF. Relative
to the *control* setup the aerosol layer has a more pronounced impact on the cloud dynamics and results in a greater decrease in
$w_e$; this is likely due to the enhanced role that evaporation of entrained air has on buoyancy production in the *05cool* setup.
Below–cloud the two setups have a consistent dynamical response, however, in the *05cool* setup the cloud maintenance is
more dependent on the below–cloud flux of water vapour. This causes a quicker decrease in BL water content which becomes
more pronounced throughout the simulation and manifests as a more pronounced period of positive SDE and a less negative
mean SDE on the third day, which in some experiments results in a positive daily mean SDE (not shown).
**3.4.2    Sensitivity to BL properties**
In the no–aerosol simulations warmer SST drives an enhanced advection of water vapour below cloud, and a lower LWP due
to an increase in BL temperature. The warmer BL also leads to stronger in–cloud buoyancy production. When the aerosol layer
is present the LWP response increases with SST, driving a stronger negative SDE in all experiments. The cloud response is
particularly sensitive to SST when the aerosol layer is near the cloud top. As discussed in Sect. 3.2, the initial response from
the weakened $w_e$, and subsequently enhanced RH, occurs quicker than the moisture source from the surface can readjust to.
The reduction in entrainment rate and BL depth are equivalent for all SST, but the greater flux of moisture from warmer SST
results in a greater increase in mean $q_t$ and RH perturbation, leading to a lower cloud base, thicker cloud, and tending to push
the SDE towards a more negative daily mean. The sensitivity of the radiative response is driven both by the SST and the
perturbation to $w_e$, therefore stronger heat perturbations closer to the cloud top result in a more pronounced sensitivity to SST.

The no–aerosol simulation for the *wetFT* setup is characterised by an LWP +5 g m$^{-2}$ greater than the *control* setup, with slightly
weaker surface evaporation. This increase in LWP is caused by entrainment of slightly moister FT air in the *wetFT* setup,
allowing the BL to maintain a greater RH. The mixing of entrained air has a smaller impact on the cloud humidity, which then
does not need to be balanced as strongly from a source at the surface. When the aerosol layer is present the weakened $w_e$
therefore has a smaller impact on the RH response of the BL, which results in a smaller SDE. This setup shows that the degree
by which the entrained air impacts the cloud plays an important role in the strength of the SDE: very dry FT air will play a
more important role in reducing RH, so that a perturbation to $w_e$ will have a greater impact on the cloud response.



**Table 5. Daily mean semi–direct radiative effect for the second day following the introduction of the absorbing aerosol layer for**
**control and cloud–sensitivity setups. All values are in daily mean Wm⁻² with increasingly negative values shaded in blue and**
**increasingly positive values shaded in red. Layer properties include the cloud–aerosol gap ('gap', in metres), the geometric thickness**
**of the layer ('dz', in metres), and the aerosol optical depth (AOD) of the layer given at a mid–band wavelength of 505 nm.**

| Type of experiment | gap | dz | AOD | *control* | *noRain* | *05cool* | *SST-1K* | *SST+1K* | *wetFT* | *800-m* | *1000-m* |
|---|---|---|---|---|---|---|---|---|---|---|---|
| Variable gap | **0** | 250 | 0.2 | -7 | -8 | -5 | -5 | -8 | -6 | 4 | 17 |
| | **100** | 250 | 0.2 | -5 | -6 | -5 | -3 | -7 | -3 | 6 | 10 |
| | **250** | 250 | 0.2 | -3 | -4 | -4 | -1 | -5 | -2 | 6 | 6 |
| | **500** | 250 | 0.2 | 0 | -1 | -2 | 1 | 0 | 0 | 4 | 2 |
| Variable thickness | 0 | **50** | 0.2 | -9 | -12 | -7 | -7 | -13 | -8 | 0 | 18 |
| | 0 | **100** | 0.2 | -8 | -10 | -7 | -5 | -11 | -7 | 2 | 20 |
| | 0 | **250** | 0.2 | -7 | -8 | -5 | -5 | -8 | -6 | 4 | 17 |
| | 0 | **500** | 0.2 | -5 | -7 | -5 | -2 | -8 | -5 | 5 | 11 |
| Variable AOD | 50 | 200 | **0.1** | -3 | -5 | -3 | -1 | -3 | -3 | 6 | 7 |
| | 50 | 200 | **0.2** | -5 | -7 | -4 | -3 | -6 | -5 | 5 | 15 |
| | 50 | 200 | **0.3** | -5 | -9 | -4 | -4 | -8 | -6 | 5 | 22 |
| | 50 | 200 | **0.4** | -6 | -9 | -5 | -4 | -10 | -5 | 6 | 25 |
| | 50 | 200 | **0.5** | -6 | -7 | -5 | -4 | -10 | -5 | 5 | 26 |


### 3.4.3 Sensitivity to BL depth

As the BL depth increases its temperature increases and the total water content decreases. Figure 9 shows the profiles of $\theta_l$ and
$q_t$ for the three setups (*control*, *800-m*, *1000-m*) during the time of strongest (0530 hours) and weakest (1300 hours) BL
dynamics. During the period with weakest dynamics the degree of coupling, or mixing, between the sub–cloud and cloud
layers is weakened. This reduces the flux of water vapour from the surface layer to the cloud, and results in a redistribution of
water from the cloud layer to the surface layer (Fig. 9b). That redistribution becomes more pronounced as the BL depth
increases, increasing BL decoupling.

Increasing the BL depth has a dramatic effect on the sign and magnitude of the SDE shown in Table 5. The SDE switches sign
from negative for a 600-m deep BL in the *control* setup to positive in the *800-m* and *1000-m* setups. The SDE in the *800-m*
setup is roughly of equal magnitude to the *control* but the *1000-m* setup is considerably greater in magnitude, peaking at
+26 Wm⁻². Responses for the base experiment shown in Fig. 10 help to understand why the BL depth has such a strong impact
on the SDE. In all setups the cloud top height decreases by ~100 m over the three days (Fig. 10a, g, and m), driven by similar
changes in $w_e$ (Fig. 10e, k, and q), however the response in cloud base height depends on the simulation and accounts for the
variation in LWP response (Fig. 10b, h, and n). In the *1000-m* setup (Fig. 10m) the cloud base decreases less than the cloud
top throughout the timeseries, driving a consistently reduced LWP.

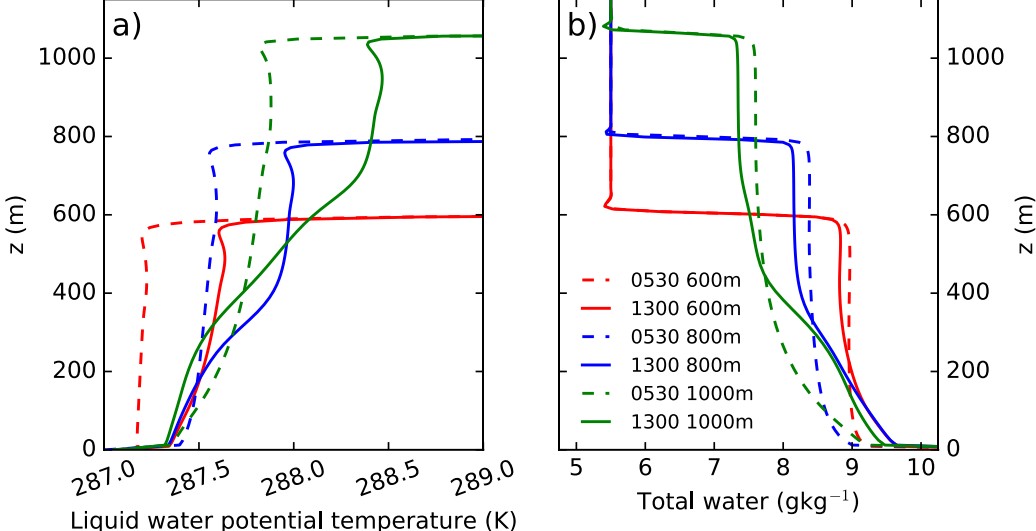


**Figure 9. Vertical profiles of a) liquid water potential temperature and b) total water mass mixing ratio taken at 0530 (dashed lines) and 1300 (solid lines) on day 1 (after spin–up) for the no–aerosol simulations.**


As shown in Fig. 9 the degree of decoupling between the sub–cloud and cloud layers increases with BL depth. The diurnal
cycle of the sub–cloud RH for the three setups (Fig. 10d, j, and p) shows that longer periods of decoupling occur as the BL
depth increases (peaks in sub–cloud RH correspond to a poorly mixed BL). In both the *control* and *800-m* setups the BL is
reasonably well mixed throughout the day. The presence of the aerosol layer enhances the midday coupling and weakens the
cloud decay phase, producing a thicker cloud in the afternoon. However, for the *1000-m* setup the lowering of the cloud layer
is not sufficient to overcome the decoupling that occurs, therefore there is no additional flux of moisture at midday and the
cloud does not thicken, producing a positive SDE in the afternoon. As the BL deepens overnight, the dynamics become
increasingly sensitive to the elevated absorbing aerosol layer (Fig. 10c, i, and o). The result is a more pronounced decrease in
the cloud growth phase overnight and a thinner cloud in the morning. The *800-m* and *1000-m* setups produce a strong positive
SDE in the morning from day 2 onwards (Fig. 10l and r), which dominates the daily mean SDE (Table 5). As described in
Sect. 3.2.2, reductions in $w_e$ and below–cloud moisture fluxes set up a feedback mechanism that decreases the BL dynamics.
As the BL deepens this mechanism occurs more rapidly and may be further enhanced by reduced cloud–top longwave cooling
that occurs when the LWP is sufficiently reduced. The reduction by ~30 g m$^{-2}$ of the LWP in the *1000-m* setup is a large
enough perturbation to reduce the longwave cloud–top cooling by ~40% and decrease buoyancy production.






**Figure 10. 3–day timeseries showing the initial response of the cloud to a 250 m thick layer of aerosol directly above the inversion with an aerosol optical depth of 0.2 from the a) – f) control setup with a boundary layer depth of 600 m, g) – l)** *800-m* **setup, and m) – r)** *1000-m* **setup.**

These results explain the different aerosol–layer sensitivities shown in Table 5. In all setups the enhanced temperature inversion weakens $w_e$ and the mixing of warm, dry FT air into the cloud layer and enhances midday coupling. For the *control* setup there is little impact on BL dynamics, so the cloud becomes thicker due to enhanced sources of moisture; as the temperature inversion strengthens this response increases. As the BL deepens the BL dynamics are increasingly weakened, driving a reduction in sub–cloud sources of moisture and a thinner cloud; as the temperature inversion strengthens this response also increases. The *1000-m* setup represents an extreme case of this scenario, whereas in the *800-m* setup the enhanced coupling is sufficient to produce an increase in sub–cloud moisture flux during the afternoon, which acts to partially mitigate the cloud thinning.



**4    Discussion and conclusions**

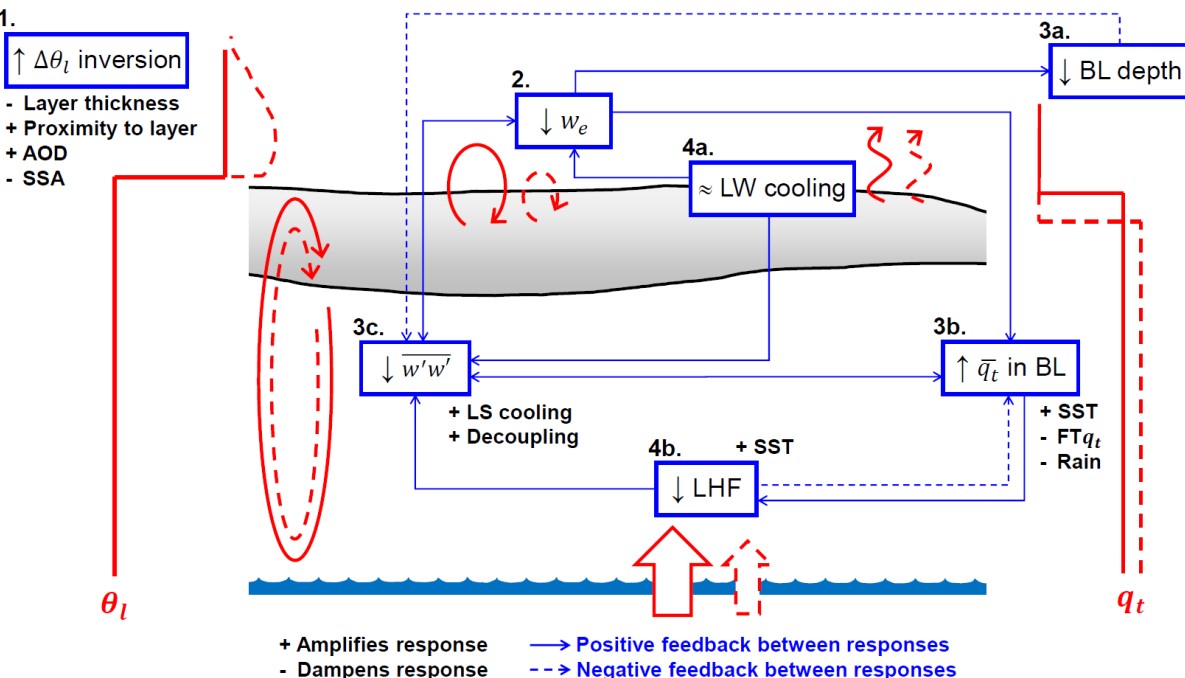


**Figure 11. Summary of how the semi–direct effect manifests in a cross section of a stratocumulus–topped boundary layer. Solid red**
**lines refer to the no–aerosol simulation and dashed red lines to the elevated absorbing aerosol–layer simulations. Key responses to**
**the boundary layer profiles are depicted in the blue boxes and include the strength of the inversion layer ($\Delta\theta_l$ inversion), entrainment**
**rate ($w_e$), boundary layer depth (BL depth), cloud–top longwave cooling (LW cooling), mean vertical motions in the boundary layer**
**($\overline{w'w'}$), mean total water content of the BL ($\overline{q_t}$), and the latent heat flux at the ocean surface (LHF). Solid (dashed) arrows between**
**boxes represent positive (negative) feedbacks between responses. For each response we include properties of the aerosol layer,**
**boundary layer, or model setup that amplify (denoted by +) or dampen (denoted by -) the response; this includes the aerosol layer**
**thickness (Layer thickness), cloud–aerosol gap (Proximity to layer), the aerosol optical depth of the layer (AOD), the single scattering**
**albedo of the aerosol layer (SSA), the sea surface temperature (SST), the water content of the free troposphere (FT$q_t$), precipitation**
**(Rain), large–scale advective heat tendency (LS cooling), and the degree of boundary layer decoupling (Decoupling).**

Figure 11 summarises the findings of this study. The SDE manifests itself as a modification to the processes that maintain the
supply of moisture to the cloud layer and are ultimately driven by the strengthened inversion layer and weakened entrainment
rate caused by an absorbing aerosol layer above the inversion. The initial sequence of responses to an elevated layer of
absorbing aerosol is summarised below, with numbers referring to each response labelled in Fig. 11:
1.   The absorbing aerosol layer produces a heat perturbation that results in a strengthened temperature inversion.
2.   Buoyant parcels of air in the BL require more energy in order to push through the strengthened temperature inversion.

627          This weakens the entrainment rate ($w_e$) across the inversion layer.

3a.  Weakened entrainment results in a decrease in the cloud top altitude and BL depth.
3b.  The reduction in the entrainment of warm and dry air from the FT reduces the amount of mixing, reducing the sink

630          of $q_t$ in the cloud layer and allowing the BL to maintain a greater RH. The result is an increase in $\overline{q_t}$ and RH.





3c. Weakened entrainment reduces the production of buoyancy from evaporative cooling of entrained air, causing a decrease in BL dynamics ($\overline{w'w'}$), especially overnight.

4a. Cloud–top longwave cooling remains unchanged due to the weak sensitivity to LWPs larger than 50 g m$^{-2}$ overnight and the relatively small changes in LWP during the daytime.

4b. Increased $\overline{q_t}$ in the BL and weakened BL dynamics reduces the evaporation rate of water from the surface, as evidenced by the reduction in latent heat flux (LHF).

According to the model sensitivity simulations presented, SDE is increased through the following mechanisms:

– Geometrically thinner aerosol layers of high aerosol density and low SSA, which produce a stronger localised heat perturbation.

– Aerosol layers close to the inversion, while larger cloud–aerosol gaps result in a delayed and weaker cloud response.

– Warmer SSTs, which enhance the flux of moisture to the BL. As a secondary response, the increased SST also drives a stronger reduction in LHF and causes the BL to adjust at a quicker rate.

Conversely, SDE is reduced by:

– Precipitation that, as a sink of cloud liquid water, dampens the cloud response. It follows that any feedbacks that result in an increase in precipitation further weakens the SDE.

– Increases to the large–scale advective heat tendency (stronger cooling), which are balanced by enhanced buoyancy production from $w_e$ and a more rapid BL adjustment.

– An increase in the moisture content of the FT, which increases the role that entrainment plays in the supply of moisture to the BL.

Finally, an increase in the degree of decoupling in the BL increases the sensitivity of the BL dynamics to changes in $w_e$, driving towards a positive daily mean SDE. Extreme cases result in a strong positive SDE from day two after applying the aerosol perturbation onwards.

Several feedbacks between responses occur as the BL adjusts to the perturbations. The key feedbacks occur in the sub–cloud layer and can work together to greatly reduce the supply of moisture to the cloud layer. Processes that act to decrease $\overline{w'w'}$ also further decrease $w_e$ and the LHF; these changes weaken the response of $\overline{q_t}$ in the BL so that there is a weaker flux of $q_v$ to the cloud layer. Reduced $w_e$ and a reduction in condensation at the base of the cloud layer weakens buoyancy production in the cloud layer which acts to further decrease $\overline{w'w'}$ and $w_e$. These feedbacks are most pronounced during the cloud growth phase overnight when the diurnal cycles of $w_e$, $\overline{w'w'}$, and LHF peak, resulting in a weakened cloud growth phase and a thinner cloud overnight and into the morning when the aerosol layer is present, thus producing a positive SDE. Longwave cloud–top cooling is only weakly sensitive to changes in LWP above 50 gm$^{-2}$ and therefore we do not see changes in the buoyancy





production from this process unless the LWP is significantly impacted, which occurs when the BL is decoupled. In this case
the reduced LWP further weakens the buoyancy production in the cloud layer, and consequently $w_e$ and BL dynamics.

A second adjustment feedback on the cloud maintenance occurs through the reduced depth of the BL which acts to promote
coupling of the cloud and sub–cloud layers. In this case the feedback mechanism outlined previously acts in reverse so that
$\overline{w'w'}$, LHF, and the supply of $q_v$ to the cloud layer increase. This weaker feedback mechanism likely occurs throughout the
diurnal cycle but only becomes important at midday when BL dynamics and sub–cloud moisture fluxes are at their weakest
and most sensitive to small changes. This adjustment results in reduced cloud decay throughout the afternoon and a thicker
cloud, and thus negative SDE, when the elevated layer of absorbing aerosol is present. The strength of this feedback mechanism
decreases as the degree of BL decoupling increases until the mechanism ceases to have any impact on the BL; in our study
this occurs when the BL is 1000 m deep.

The sign and magnitude of the SDE from elevated layers of absorbing aerosol is sensitive to the layer properties and BL
properties, especially the diurnal variations in coupling between the cloud and sub–cloud layers. For coupled BLs, the SDE on
the first day after adding the absorbing aerosol layer is slightly positive unless the aerosol layer is close to the inversion layer.
On the second and third day the SDE is strongly negative and peaks on the second day. Generally, for coupled BLs the SDE
is of opposite sign to the DRE and often greater in magnitude, resulting in a small or negative total radiative effect for aerosol–
radiation interactions from elevated absorbing aerosol layers. For BLs that show characteristics of being decoupled for most
of the diurnal cycle the SDE is positive for all three days and increases in magnitude throughout; as the BL becomes more
decoupled the magnitude of the SDE increases. For decoupled BLs the SDE acts to enhance the DRE, resulting in a larger total
radiative effect.

The increased LWP and negative SDE in the well–mixed coupled BL experiments is consistent with satellite observations over
the southeast Atlantic from Adebiyi and Zuidema (2018) and Wilcox (2012). However, our LEM simulations suggest a positive
SDE in decoupled BL regions, such as near the stratocumulus–to–cumulus transition region. In reality, the BL may not be as
decoupled as in the simulations. The deepening BL is usually accompanied by an increasing SST (Sandu and Stevens, 2011)
which was not represented in our simulations; the increase in SST would provide a considerably larger flux of moisture from
the surface and enhance the production of buoyancy at the surface which may act to weaken the sensitivity of the BL to changes
in dynamics. Contrary to the results presented here, the stratocumulus–to–cumulus transition LES studies by Yamaguchi et al.
(2015) and Zhou et al. (2017) suggest that only those elevated smoke layers that are very close, or in direct contact with, the
cloud layer impact the cloud properties. However, in these studies the prescribed subsidence rate above the cloud layer was -1.5
to -2 mms$^{-1}$, which is lower than used in our study (-5 mms$^{-1}$ at an equivalent altitude) and would delay the response from the
heat perturbation. This difference in subsidence rate represents an important sensitivity to the impact that elevated layers may
have on the cloud, both in terms of LES and in the real–world. It is worth noting that Yamaguchi et al. (2015) and Zhou et al.





(2017) used the same case study (Sandu and Stevens, 2011) yet found opposing results on whether the absorbing aerosol layer
inhibits or hastens the transition to cumulus. Yamaguchi et al. (2015) state that throughout their simulations the BL is decoupled
below 800 m, whereas in Zhou et al. (2017) vertical mixing within the BL continues until the inversion height exceeds ~1.4
km (Zhou et al., 2017; Fig. 1b). Our results highlight that the cloud response is sensitive to the diurnal variations in BL mixing,
which may explain these opposing results.

Satellite products provide an excellent opportunity to observe aerosol–cloud and aerosol–radiation interactions in remote
locations such as the southeast Atlantic Ocean, however most instruments are on polar orbiting satellites that only provide
observations from a limited window within the diurnal cycle of the clouds. Our simulations suggest the cloud response to
elevated absorbing aerosol layers and the SDE display important diurnal variations so a single observation is unlikely to be
representative of the daily mean response. Important changes to the cloud properties occur overnight and play a considerable
role in the SDE of the morning period, yet little is known about the impact from absorbing aerosol layers overnight. Future
studies should use geostationary satellite observations to investigate the full diurnal cycle of the SDE.

For a well–mixed coupled BL, the initial cloud and radiative response depend on small–scale processes, such as entrainment
and turbulence, which must to be parameterised in climate models. Gordon et al. (2018) used a nested regional model within
the Hadley Centre Global Environment Model (HadGEM) to investigate the impact of an incoming elevated plume of smoke
in the southeast Atlantic. They found that the elevated aerosol layer reduced cloud top height and enhanced LWP through a
reduction in $w_e$ driven by localised heating at or just above the cloud layer of ~6 K. The importance of the weakened $w_e$ aligns
well with the LES results of the present study, but the magnitude of the cloud and radiative response are much greater in
HadGEM, with an LWP increase of 90%, an increase in cloud fraction of 19% and a mean SDE of -30 Wm$^{-2}$. Gordon et al.
(2018) do not find a consistent longer–term (~3 days) reduction in LWP following BL adjustments. In the LES simulations
presented here, cloud fraction remained ~100%, which may the smaller SDE than Gordon et al. (2018). Additionally,
concurrent aerosol-cloud interactions may modify the underlying cloud properties, which may act to amplify the SDE. The
lack of BL adjustment may be due to processes that are not explicitly treated in HadGEM, such as BL turbulence and surface
fluxes, or due to aerosol-cloud interactions not represented in the LES. Alternatively, differences may be due to different
simulated cases. The trajectory analysis of Gordon et al. (2018) suggests that their BL air mass traverses the study region more
quickly than the absorbing aerosol layer, which may prevent the BL adjustments from occurring.

In our simulations the SST and subsidence rate are held constant for the whole duration whereas real stratocumulus decks tend
to experience an increasing SST and decreasing subsidence rate. An increasing SST increases surface latent heat fluxes, cloud
liquid water content, and the strength of BL eddies, and acts to deepen the BL through increased entrainment and enhance
decoupling of the sub–cloud layer (Bretherton and Wyant, 1997). As the cloud is advected over the warmer sea surface the
enhanced flux of moisture would act to increase the magnitude of the SDE and prevent the BL from drying out as quickly.





Simultaneously, the enhanced decoupling of the sub–cloud layer may result in BL dynamical feedbacks that result in a
reduction in LWP (see Fig. 10). Changes to the aerosol distribution within the cloud or in the cloud droplet distribution have
not been considered in this study. A weakened $w_e$ increases condensate in the cloud and likely results in an increase in cloud
droplet effective radius ($r_e$). This would promote warm rain process and enhance precipitation, thus reducing the LWP and
amplifying the reduction in BL dynamics. These combined effects could lead to a decrease in LWP and shift the SDE towards
a positive sign at a quicker rate than suggested by the LES. For the cases where the aerosol layer is directly above the smoke
layer an enhanced flux of CCN into the BL would be expected and would act to reduce $r_e$, supress precipitation, and act to
enhance buoyancy production. The introduction of the absorbing aerosol into the cloud layer would additionally enhance cloud
evaporation and act to thin the cloud layer (Hill and Dobbie, 2008; Johnson et al., 2004). Thus, although the experiments where
the aerosol layer is directly above the inversion result in the most strongly negative SDE, the response would be at least
partially mitigated if the aerosol distribution was represented explicitly. Extending the present study using a binned
microphysics scheme would include the additional response of the droplet size distribution and using an aerosol scheme would
include the additional impacts the weakened $w_e$ has on the availability of CCN and subsequent cloud response.
**5    Appendix**
This appendix describes how the AOD and SSA is prescribed in elevated aerosol layer experiments, along with the geometric
thickness of the aerosol layer and the distance between the inversion layer and the aerosol base. In each call to the radiation
scheme the desired AOD and SSA are used to determine the mass mixing ratio of two aerosol species, water–soluble like (WS)
and biomass–burning like (BB).

For a single wavelength, the AOD between the altitudes $z_0$ and $z$, corresponding to the base and top of the aerosol layer
respectively, is calculated as:

$$\text{AOD} = \sum_{i=z_0}^{z} \sum_{j=WS,BB} \left( K_{scat\,j} + K_{abs\,j} \right) \cdot q_{i,j} \cdot \rho_i \cdot dz_i \qquad \text{(A1)}$$


where $K_{scat}$ and $K_{abs}$ are the specific scattering and absorption coefficients, respectively, for the aerosol species $j$, in units
$m^2\,kg^{-1}$, with mass mixing ratio $q$ in kg kg$_{dry}^{-1}$, at each model level $i$ of geometric thickness $dz$ in m, and density of dry air $\rho$ in
kg m$^{-3}$. If the mass mixing ratio of each species is assumed equal and constant with height ($q_{WS} = q_{BB}$ and $q_i = q$), Eq. A1
becomes:

$$q \cdot \sum_{i=z_0}^{z} \rho_i \cdot dz_i = \frac{\text{AOD}}{\sum_{j=WS,BB} K_{scat\,j} + K_{abs\,j}} \qquad \text{(A2)}$$




We incorporate a factor $X_{SSA}$ into Eq. A2 that can be used to describe the relative ratio of WS mass to BB mass so that Eq. A2
becomes:

$$q \cdot \sum_{i=z_0}^{z} \rho_i \cdot dz_i = \frac{\text{AOD}}{\left(K_{scat_{WS}} + K_{abs_{WS}}\right) + X_{SSA} \cdot \left(K_{scat_{BB}} + K_{abs_{BB}}\right)} \tag{A3}$$


Equation A3 can be re-arranged to give $q$ for a given AOD:

$$q = \frac{\text{AOD}_{constant}}{\sum_{i=z_0}^{z} \rho_i \cdot dz_i} \tag{A4}$$

where

$$\text{AOD}_{constant} = \frac{\text{AOD}}{\left(K_{scat_{WS}} + K_{abs_{WS}}\right) + X_{SSA} \cdot \left(K_{scat_{BB}} + K_{abs_{BB}}\right)} \tag{A5}$$


Therefore for the two aerosol species:

$$q_j = \begin{cases} q, & j = WS \\ X_{SSA} \cdot q, & j = BB \end{cases} \tag{A6}$$


The overall SSA is calculated as:

$$\text{SSA} = \frac{K_{scat_{WS}} + X_{SSA} \cdot K_{scat_{BB}}}{K_{scat_{WS}} + X_{SSA} \cdot K_{scat_{BB}} + K_{abs_{WS}} + X_{SSA} \cdot K_{abs_{BB}}} \tag{A7}$$


Equation A7 can be re-arranged to solve for $X_{SSA}$

$$X_{SSA} = \frac{K_{scat_{WS}} - \text{SSA} \cdot \left(K_{scat_{WS}} + K_{abs_{WS}}\right)}{\text{SSA} \cdot \left(K_{scat_{BB}} + K_{abs_{BB}}\right) - K_{scat_{BB}}} \tag{A8}$$


At the beginning of the simulation $X_{SSA}$ and $\text{AOD}_{constant}$ are calculated using Equations A8 and A5, respectively, using the
shortwave extinction coefficients of the aerosols for the wavelength band 320 – 690 nm and the prescribed AOD and SSA. At
each horizontal grid point $q$ is then calculated using Eq. A4 for the elevated aerosol layer where $z_0$ is the base of the aerosol
layer, and $z$ is the top of the aerosol layer. The mass mixing ratio of each species is calculated using Eq. A6 and finally the
mass mixing ratio profiles of WS and BB applied to the radiation scheme.



## 6 Author contribution

RJH, NB, EJH, and AAH designed the methodology and experiments. AAH provided model expertise and assistance. RJH setup, performed, and post-processed the simulations. RJH, NB, EJH, and AAH analysed the results. RJH provided all visualisations and wrote the initial manuscript draft. NB, EJH, and AAH provided revisions and commentary on the manuscript.

## 7 Competing interests

The authors declare that they have no conflict of interest.

## 8 Acknowledgments

This research was funded by the UK Natural Environment Research Council (NERC) CLouds and Aerosol Radiative Impacts and Forcing: Year 2016 (CLARIFY-2016) project NE/L013479/1. We acknowledge use of the Monsoon system, a collaborative facility supplied under the Joint Weather and Climate Research Programme, a strategic partnership between the Met Office and the Natural Environment Research Council. The CALIOP data were obtained from the NASA Langley Research Center Atmospheric Science Data Center.

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
