# Peer review of "Diurnal cycle of the semi-direct effect from a persistent absorbing aerosol layer over marine stratocumulus in large-eddy simulations"

_Atmospheric Chemistry and Physics, 2019_

## Referee Comment (RC1) · Anonymous Referee #2 · 25 Jun 2019

**General remark**

The paper discusses in depth the dynamics of the stratocumulus-topped subtropical marine boundary layer and its sensitivity to an elevated absorbing aerosol layer using a sophisticated LES model. The model experiment is well designed, but could benefit from a more detailed representation of cloud microphysics including aerosol-cloud interactions, which in fact was discussed. On the other hand, more degrees in freedom would certainly complicate interpretation of model results. The sensitivity study was carried out in a well-structured fashion in order to distinguish the effects of variable aerosol-layer properties on the cloud layer. As such, the type and number of experiments seems reasonable to cover the wide range of possible scenarios. It is further

laudable the extension of the sensitivity runs to cover the influence of meteorological and model parameters to corroborate the primary results. The study highlights the complexity and variability of the semi-direct effect of aerosols on cloud decks, which are too often simplified in climate models. As such, it gives a clear and consistent picture on how the variation of the aerosol layer properties affect the sign and magnitude of the semi-direct effect. Different environmental variables dampen or enhance the cloud response. The timing of the interaction process was found to be crucial, as the boundary layer evolves diurnally and eventually adjusts to the external forcing, which potentially reverses the sign of the semi-direct effect from an initial growth of the cloud. Eventually, the authors used the benefits of a LES model to give a thorough analysis of the altered thermodynamic properties and dynamics within the boundary layer communicating the feedback mechanisms, which is an important and interesting contribution to the topic. The well-structured paper contains an appropriate number of plots to visualize the data in order to promote the readers understanding and traceability of the text content. Concluding from the above, the paper presents a valuable contribution to ACP.

**Minor comments**

Page 4, line 74: Plural "extend of cloud-aerosol gaps"

Page 5, Model setup: Maybe add a sentence to the lateral boundary conditions. I assume they are periodic?

Page 6 and 7, Setup of elevated-aerosol experiments: Unfortunately, the CALIOP measurements are not that reliable, which makes this paragraph less significant. For cloud measurements it is often a tradeoff between accuracy and representativeness of different datasets. Aren't there other data available, like aircraft measurements, that could complement the used data?

[Figure]

Page 7, line 195: I refer to "This type of experiment is analogous to a satellite retrieval that estimates the AOD and aerosol layer top but does not detect the lower extend of the aerosol layer." How can this be analogous, if you cannot infer the geometric thickness? Do you assume an extinction profile?

Page 7, end of line 185: "absorbing aerosol" instead of "layer".

Page 8, Eq.1: The ordering of the flux terms in the formula is wrong. It must be:

$$SDE = F_{TOA,aerosol} - F_{TOA,no-aerosol} - DRE \tag{1}$$

Page 14, Fig. 5: Is below-cloud RH the vertical mean for the distance from ocean surface to the cloud base? Otherwise, at which height is the value taken?

Page 15, line 340: Based on Fig. 6e the total water path (TWP) (units kgm-2), and not the total water content (TWC) (kgm-3) is compared. The reduction in total water path is in-line with the reduced BL height (which also decreases by about 15

Page 17, Eq. 3: This equation is confusing. The following formulation should be equivalent:

$$\begin{aligned} Z_{lower} &= Z_{max} \cdot (1 - 0.025) \\ Z_{upper} &= Z_{max} \cdot (1 + 0.25) \end{aligned} \tag{2}$$

If not, please rewrite it in a more understandable way.

Page 17, line 374: The explanation provided is not very convincing. Isn't the initial peak of positive SDE occurring before or around midday? Anyway, at the time it occurs, the clout top height and entrainment rate seem not to be significantly affected by the aerosol layer (look at Fig. 5 or Fig. 7 red line of the 500m-gap experiment). How much does the elevated aerosol layer affect radiative cooling of the cloud tops at night? Does the initial positive spike in SDE could be related to this?

Page 19, lines 415 - 417: See comment above.

Page 27, Fig. 10 e,k,q: Please specify "BL mean" and "BL total". I assume the dotted line is "BL total"? If "BL total" refers to TWP and "BL mean" to TWC you can see how the moisture content of air increases within the BL, despite an overall decrease in TWP due to the shrinking of the BL.

Page 29, line 633: This can only hold true if it is reasonable to neglect emission of longwave radiation of the aerosol layer and ergo its insulating effect.

Page 29, line 638: ", the magnitude of SDE is increased . . .", or ", SDE is amplified . . ." is less ambiguous, as the sign of SDE is negative.

Page 31, line 719: missing "explain"
* * *

---

## Referee Comment (RC2) · Anonymous Referee #1 · 26 Jun 2019

This manuscript studies the semi-direct effect on marine stratocumulus clouds with large eddy simulations. Due to their rather simpler model (e.g., one moment microphysics, fixed smoke layer) compared with the recent LES studies investigating the interactions between clouds and biomass burning smoke, this study has to exclude the indirect effects and focuses only on the semi-direct effect. For this perspective, the argument for the steady state response is irrelevant to the reality, even though their results answer the cloud response when there is no indirect effect. I think that this study is far from complete for their objectives. In order to study how much the semi-direct effect modulates stratocumulus, one has to include the indirect effect and then quantify these effects. There are small scientific progresses and understandings in the absence of the indirect effects, but I do not see significant advances from e.g., Hill and Dobbie

(2008) and Johnson et al (2004).

There are a few model configuration issues. The model configuration is in the Eulerian framework but the heated layer descends due to subsidence while the smoke layer does not. If the smoke layer is assumed to be at a constant height due to large scale horizontal transport, then the heated layer should follow it. If the model is in the Lagrangian framework, then both smoke layer and heated layer should be transported by subsidence. In either case, the indirect effects should be considered when the smoke layer is touched to the boundary layer top (their figures show positive entrainment rate). I also found difficulties to imagine that with cloud droplet number of 240 cm-3, well mixed, 600 m depth PBL, and LWP of around 60 gm-2, the stratocumulus produces precipitation that significantly alters results compared with the noRain case. Something may be wrong in their model. I do not think that their model is suitable to study their objectives in the present. They should spend some time to modernize their model.

For this reason, I reluctantly reject the manuscript and encourage for resubmission.

---

## Author Comment (AC1) · 23 Aug 2019

**Response to reviewer #2**

We thank the reviewer for their helpful comments and the suggested changes / clarifications. We address each comment in turn below. Reviewer comments are in bold, and changes made to the manuscript in italics.

**Page 4, line 74: Plural "extend of cloud-aerosol gaps"**

The manuscript has been amended to read "*extent of cloud-aerosol gaps*".

**Page 5, Model setup: Maybe add a sentence to the lateral boundary conditions. In assume they are periodic?**

The reviewer is correct; the lateral boundary conditions are periodic. This has been included in the model setup section.

**Page 6 and 7, Setup of elevated-aerosol experiments: Unfortunately, the CALIOP measurements are not that reliable, which makes this paragraph less significant. For cloud measurements it is often a trade-off between accuracy and representativeness of different datasets. Aren't there other data available, like aircraft measurements that could complement the used data?**

We have complemented the CALIOP (532 nm channel) analysis with the Cloud-Aerosol Transport System (CATS) 1064-nm lidar dataset on-board the International Space Station (CATS-ISS_L2O_D-M7.2-V3-00_05kmLay). The CATS lidar retrieves feature altitudes using the 1064 nm wavelength channel, which is better able to retrieve the lower extent of the aerosol layer than the 532 nm channel used by CALIOP as it is not fully attenuated (Rajapakshe et al., 2017). The CATS dataset is only available for 3 years, compared to the 10 years of CALIOP data used for the current climatology, hence we present both datasets. Figure R1 below shows that the aerosol layer is closer to the cloud top in the CATS dataset than in the CALIOP dataset. Both datasets display considerably variability in the cloud-aerosol gap but provide evidence that the gap is most likely less than 1000 m. Therefore our experimental design remains appropriate. The corresponding figure in the manuscript (Figure 1) has been updated with Figure R1 and the text has been updated to reflect these changes.

[Figure]

*Figure R1. Normalised frequency of occurrence of gap distance between cloud layer top and aerosol base heights from CALIOP (blue solid line) and CATS (red dotted line) for single layer coincidences of aerosol and cloud in the months of July, August, and September (2007–2016 for CALIOP; 2015-2017 for CATS) over the southeast Atlantic (15°S to 2.5°N, 10°W to 10°E). Gaps are binned from -1.5 to 5.5 km in 200 m increments and data in each grid has been normalised to the maximum frequency across the whole study area. The percentage of scenes where the aerosol layer base is less than 360 m above the cloud top height is shown in the top right of each subplot, in blue for CALIOP and red for CATS.*

**Page 7, line 195: I refer to "This type of experiment is analogous to a satellite retrieval that estimates the AOD and aerosol layer top but does not detect the lower extend of the aerosol layer." How can this be analogous, if you cannot infer the geometric thickness? Do you assume an extinction profile?**

Our explanation was confusing and has been clarified. The situation we are describing is when column-integrated total AOD, as retrieved for example by MODIS, is combined with partial knowledge of the layer geometric thickness, as retrieved for example by CALIOP. This combination occurs for example in the CCCM product. We have rewritten the paragraph to clarify the point we are making as follows:

"*This type of experiment aims to understand the importance of correctly retrieving the full extent of the aerosol layer from a satellite retrieval when only the AOD is known. Those variables are for example provided in the combined CCCM satellite product (Kato et al., 2010; 2011).*"

**Page 7, end of line 185: "absorbing aerosol" instead of "layer".**

This has been amended as suggested.

**Page 8, Eq.1: The ordering of the flux terms in the formula is wrong. It must be:**

$$SDE = F_{TOA,aerosol} - F_{TOA,no-aerosol} - DRE$$

Agreed. We have corrected the formula in the manuscript.

**Page 14, Fig. 5: Is below-cloud RH the vertical mean for the distance from ocean surface to the cloud base? Otherwise, at which height is the value taken?**

The below-cloud RH is the vertical mean from above the ocean surface to the cloud base. We have clarified this in the manuscript and updated Figure 5 and Figure 6. While amending the figures, a mistake in plotting RH at the surface layer has been detected. This has been fixed and the mistake does not affect any of the text as the same response is observed in all figures.

**Page 15, line 340: Based on Fig. 6e the total water path (TWP) (units kgm-2), and not the total water content (TWC) (kgm-3) is compared. The reduction in total water path is in-line with the reduced BL height (which also decreases by about 15**

We have amended the manuscript to read '*total water path*' and have changed the variable 'Total BL qt' to '*TWP of BL*' throughout the manuscript. The reviewer's comment seems to have been truncated, so we cannot respond to the missing part.

**Page 17, Eq. 3: This equation is confusing. The following formulation should be equivalent:**

$$Z_{lower} = Z_{max} \cdot (1 - 0.025)$$
$$Z_{upper} = Z_{max} \cdot (1 + 0.25)$$

**If not, please rewrite it in a more understandable way.**

Agreed. The suggested equation is more understandable. We have rewritten the equation as suggested.

**Page 17, line 374: The explanation provided is not very convincing. Isn't the initial peak of positive SDE occurring before or around midday? Anyway, at the time it occurs, the clout top height and entrainment rate seem not to be significantly affected by the aerosol layer (look at Fig. 5 or Fig. 7 red line of the 500m-gap experiment). How much does the elevated aerosol layer affect radiative cooling of the cloud tops at night? Does the initial positive spike in SDE could be related to this?**

**Page 19, lines 415 - 417: See comment above**

As the reviewer rightly suggests, the positive SDE occurs before midday rather than after midday as we erroneously wrote. This has been corrected.

The reviewer is not convinced by our explanation and suggests the positive SDE may be influenced by changes to the cloud-top longwave cooling. We have addressed this comment by including a new figure (Figure R2 in this document, Figure S1 in the revised manuscript) in the supporting information and improving our explanation in the manuscript. The new figure focuses on the cloud response in the first day (from 0230 to 1600) and includes the changes to cloud properties, buoyancy flux, advected total water content tendency, cloud-top longwave cooling, and LW fluxes for three of the experiments with a variable cloud-aerosol gap (with AOD=0.2 and layer geometric thickness of 250m).

The new figure shows that the positive SDE is driven by the decrease in LWP (Figure R2b) that is most evident at 0830 and 1000 for the experiments with gaps of 0 and 100 m. This response is caused by an increase in cloud base height (Figure R2a) without a corresponding change in cloud top height, which thins the cloud and reduces the LWP. Cloud base height increases because of weaker mixing within the boundary layer which reduces the transport of moisture within and beneath the cloud (Figure R2d). That reduced transport occurs because at 0830 the buoyancy flux throughout the profile weakens (Figure R2c), at the time at which entrainment starts to sharply decrease (Figure 5b). Note that below-cloud RH (Figure 5d) does not increase until after midday which indeed suggests that the increasing cloud base height is driven by in-cloud changes or the flux of moisture to the cloud base. As the day progresses the continued reduction in entrainment rate results in a moister boundary layer and an increase in RH below the cloud, which allows the cloud base to decrease and LWP to increase. This explains why stronger perturbations to the entrainment rate on the first day (such as when the layer is close to the cloud) results in a quicker recovery of the LWP (see Figure 7). This improved explanation has been included in Section 3.3.1.

With regards to the cloud-top LW cooling, we would expect the cooling rate to be weakened by the presence of the elevated aerosol layer and any additional heating of the layer, both of which would increase downwelling LW. Figure R2e shows that there are instantaneous differences in LW cooling up to a magnitude of 20 K day$^{-1}$, however the sign changes throughout the day. The response of the net LW flux shown in Figure R2f confirms that there is little impact to the fluxes above cloud before sunrise. The buoyancy flux profiles in Figure R2c do show a limited response to the aerosol layer before sunrise (at 0400 hours) in all experiments but there is little simultaneous LWP response. The dominant cloud response appears to occur after sunrise, which suggests the decrease in LWP is driven by an enhanced inversion strength rather than weakened cloud-top LW cooling. We have included a brief discussion of this additional effect in Section 3.2.1.

[Figure]

Figure R2. Response to the presence of an aerosol layer above the cloud (gap of 0 m in blue, 100 m in red, and 500 m in green) of a) the cloud top (solid line) and cloud base (dashed line) heights, b) the cloud liquid water path (LWP), c) profiles of the mean buoyancy flux, d) profiles of the mean advected total water content tendency, e) cloud-top longwave cooling, and f) profiles of mean longwave net flux (positive values indicate increased downward flux). The geometric thickness of the aerosol layer is 250 m and its optical depth is 0.2. Data is shown for the first day following the introduction of the aerosol layer. Mean instantaneous profiles (shown in panels c, d, and f) for each time are centred on a value of zero, depicted by the vertical dotted lines. Each profile is separated on the x-axis by a constant magnitude shown above each corresponding plot.

**Page 27, Fig. 10 e,k,q: Please specify "BL mean" and "BL total". I assume the dotted line is "BL total"? If "BL total" refers to TWP and "BL mean" to TWC you can see how the moisture content of air increases within the BL, despite an overall decrease in TWP due to the shrinking of the BL**

We have amended the figure and text as suggested to provide clarity.

**Page 29, line 633: This can only hold true if it is reasonable to neglect emission of longwave radiation of the aerosol layer and ergo its insulating effect**

This relates to the comment about cloud-top LW cooling addressed above with Figure R2. Figure R2f shows the response of the net LW flux profiles. Before sunrise the net fluxes above cloud are < 1.5 % greater when the aerosol layer is present, suggesting a weak insulating effect. During the day this increases up to a maximum of ~ 5 % as the temperature of the aerosol layer increases, but still indicates a weak insulating effect.

Section 3.2.1 already contains a discussion of the changes to LW fluxes and cloud-top cooling following the comment above, and we have added the following sentence to the discussion and conclusions section:

"*The insulating effect of the aerosol layer only weakly influences the net longwave fluxes and divergence above the cloud.*"

**Page 29, line 638: ", the magnitude of SDE is increased. . .", or ", SDE is amplified. . ."is less ambiguous, as the sign of SDE is negative.**

Agreed. The manuscript has been amended to read "*SDE is amplified*".

**Page 31, line 719: missing "explain"**

The manuscript has been amended as suggested.

**References**

Kato, S., S. Sun-Mack, W. F. Miller, F. G. Rose, Y. Chen, P. Minnis, and B. A. Wielicki, 2010: Relationships among cloud occurrence frequency, overlap and effective thickness derived from CALIPSO and CloudSat merged vertical profiles. J. Geophys. Res., 115, D00H28, https://doi.org/10.1029/2009JD012277

Kato, S., and Coauthors, 2011: Improvements of top-of-atmosphere and surface irradiance computations with CALIPSO-, CloudSat-, and MODIS-derived cloud and aerosol properties. J. Geophys. Res., 116, D19209, https://doi.org/10.1029/2011JD016050

Rajapakshe, C., Zhang, Z., Yorks, J. E., Yu, H., Tan, Q., Meyer, K., Platnick, S. and Winker, D. M. 2017: Seasonally transported aerosol layers over southeast Atlantic are closer to underlying clouds than previously reported, Geophys. Res. Lett., 44(11), 5818–5825, doi:10.1002/2017GL073559

---

## Author Comment (AC2) · 23 Aug 2019

**Response to reviewer #1**

We thank the reviewer for their time in evaluating our paper and we thank the reviewer for their frank comments. We respectfully disagree with their conclusions. However, the reviewer highlights some weaknesses which we have addressed in the manuscript. The reviewers' criticisms are:

1. That the study only offers small advances over current literature;
2. That the model used is not state of the art;
3. That the experiment configuration is unrealistic.

In the following, we answer those points in turn to stress that:

1. Our study provides novel insights into semi-direct effect processes that suggest they are much more subtle and more elusive than previously thought;
2. The model used in our study contains the appropriate amount of complexity to answer our scientific objective, and that adding complexity would unnecessarily complicate the picture;
3. The experiments that we performed are relevant to understanding real semi-direct perturbations of marine stratocumulus clouds.

These points are reflected in the changes we have made to the revised manuscript to address these arguments.

1. The first major criticism is that this study offers only small advances over previous studies. To date there are only two high-profile reviews of black carbon (BC) semi-direct effect on clouds: Koch and Del Genio (2010) and Bond et al. (2013), which addresses BC impacts on climate more generally. Both reviews rely on just a single high resolution modelling study (Johnson et al., 2004) and a single case to support the conclusion that the semi-direct effect is negative on a global average. More recent studies have focused on stratocumulus-to-cumulus transition (Yamaguchi et al., 2015; Zhou et al., 2017), which makes isolating semi-direct effects difficult. Given this paucity in model and observations, the role of BC over marine stratocumulus is still a major uncertainty. Observations have shown that BC can occur at various heights above marine stratocumulus and this study investigates how this variability translates into the vertical profile of heating produced and the response of the underlying cloud.

   Our results strongly suggest that semi-direct effects are much more subtle than previous literature assessed. We find that the semi-direct effect strongly weakens with increasing gap between cloud and BC layer – as soon as the gap in larger than about 100 m, semi-direct effects become unimportant. We also find a strong diurnal cycle that means that, although semi-direct effects may be large instantaneously, changes of signs with time provide a weak daily average. To our knowledge these conclusions are entirely new and build upon the very small collection of high-resolution modelling studies that have studied the semi-direct effect of elevated BC layers above stratocumulus. If the reviewer is aware of studies that we have missed we would greatly welcome these references.

   Considering the reviewer's criticism, it is apparent that the importance of the main results was not made clear enough. We have therefore amended the abstract and conclusions in the revised manuscript to make this clearer and to highlight the new results.

2. We agree that the model used in this work is microphysically simpler than other recent studies (e.g., Yamaguchi et al., 2015; Zhou et al., 2017). The microphysics in this work is single moment, while the other studies use double moment scheme with impacts on the cloud droplet distribution. While such a simplification would be problematic if we were investigating the interaction of BC aerosols with the cloud, in this work, we instead focus on BC above the cloud. We agree that the issue is simplified by not advecting / subsiding BC aerosols but this type of set-up matches the scientific objective of assessing the impact of aerosol layers above the cloud. The Large Eddy Model (LEM) has a long track record of being used to study cloud-precipitation-aerosol interactions for several cloud regimes and was included in several LES inter-comparisons. To cite only the studies

published in the past 10 years: Hill et al., 2009, Hill et al., 2014, Efstathiou et al., 2015; Efstathiou et al., 2016; Ackerman et al., 2009; Dussen et al., 2013; Ovchinnikov et al., 2014; De Roode et al., 2016). We have added a sentence into the model description section to demonstrate the track record of the LEM.

As recognised by Reviewer #2 we designed our experiments to study the semi-direct effect from the bottom–up with a systematic approach that allowed us to investigate, for the first time, the sensitivity of the thermodynamic response of the boundary layer to properties of the BC layer, as well as the meteorological conditions and key model parameters. In that context, complexity needs to be added where it is useful. The reviewer mentions two specific limitations of our model: the lack of representation of aerosol indirect effects, and the Eulerian framework used by the model.

Our model does not consider indirect effects because they would quickly muddy the water. We however agree that potential mitigating impacts of indirect effects need discussing and have expanded our current discussion on possible impacts from indirect effects in Section 4 of the revised manuscript. As discussed by Petters et al. (2012) some modelling studies focusing on stratocumulus find that LWP decreases with cloud droplet number concentration (Nd), whereas others find that LWP increases. Some studies find increases in entrainment rate, whereas others show decreases. The diversity in response was attributed to differences in modelling frameworks and the profiles of state variables used to initialise the model. In addition, in-situ observations routinely find that the BC over the Southeast Atlantic is transported in moist layers. As the BC layer is entrained into the cloud layer the increased flux of water from the free-troposphere could act to mitigate the changes in LWP and entrainment that occurs alongside an increased Nd.

Our model uses a Eulerian framework where the BC layer remains at a constant height above the cloud whereas the heat perturbation is allowed to subside into the cloud. Although we agree that in reality both should subside, the sensitivity experiments that form the core of our study include changes to the gap between cloud and BC layer. Therefore we learn from our model that if the BC layer could subside with the heat, an enhancement of the inversion strengthening would be seen. We agree that this point should be added to the discussion, and have included this in Section 4, but stress that it does not affect our conclusions and the novelty of the study.

The reviewer believes there may be something wrong with our model due to the large impact that precipitation has on our results. The cloud-base precipitation rate obtained in our model configuration ranges from 0.2 mm day$^{-1}$ at night to 0.01 mm day$^{-1}$ during the day. For a cloud with a LWP of 60 g m$^{-2}$ this is within the range of observations presented by Abel et al. (2010). As discussed in Ackerman et al. (2009) and Wood et al. (2012), drizzle plays an important role in the dynamical processes throughout the boundary layer, therefore we do not believe this aspect of our results is wrong. A brief evaluation of the precipitation rate has been included in Section 3.1 of the revised manuscript.

3. We agree with the reviewer that the steady state stage of our simulations is not realistic. Indeed, that is acknowledged in the text. But the core of the paper, which provides the novel results, focuses on the initial response, which is realistic. The steady state response is not analysed at all for the sensitivity experiments. We have added text to Section 3.2 to clarify the reason for the simulations.

In summary, we stand by our model, its setup, and the range of experiments that we performed. Our bottom–up approach allows us to robustly study the semi–direct effect and test considerably more parameter space than previous studies. Our results build upon a very small collection of modelling studies and provide the community with much needed insight into the subtleties of semi-direct responses of stratocumulus clouds. But we thank the reviewer for highlighting shortcomings in the description of our work, which we have addressed in the revised version.

**References:**

Abel et al., 2010, 'Evaluation of stratocumulus cloud prediction in the Met Office forecast model during VOCALS-Rex', Atmos. Chem. Phys., doi:10.5194/acp-10-10541-2010

Ackerman et al., 2009, 'Large-Eddy Simulations of a Drizzling, Stratocumulus-Topped Marine Boundary Layer', Mon. Weather Rev., doi:10.1175/2008MWR2582.1

Bond et al., 2013, 'Bounding the role of black carbon in the climate system: A scientific assessment', J. Geophys. Res. Atmos., doi:10.1002/jgrd.50171

De Roode et al., 2016, 'Large-Eddy Simulations of EUCLIPSE–GASS Lagrangian Stratocumulus-to-Cumulus Transitions: Mean State, Turbulence, and Decoupling', J. Atmos. Sci., doi: 10.1175/JAS-D-15-0215.1

Dussen et al, 2013, 'The GASS/EUCLIPSE model intercomparison of the stratocumulus transition as observed during ASTEX: LES results', J. Adv. Model. Earth Syst., 5, 483– 499, doi:10.1002/jame.20033.

Efstathiou and Beare, 2015, 'Quantifying and improving sub-grid diffusion in the boundary-layer grey zone' Q.J.R. Meteorol. Soc., 141: 3006-3017. doi:10.1002/qj.2585

Efstathiou et al., 2016, 'Grey zone simulations of the morning convective boundary layer development', J. Geophys. Res. Atmos., 121, 4769– 4782, doi:10.1002/2016JD024860.

Johnson et al., 2004, 'The semi-direct aerosol effect: Impact of absorbing aerosols on marine stratocumulus', Q. J. R. Meteorol. Soc., doi:10.1256/qj.03.61

Hill et al., 2009, 'The Influence of Entrainment and Mixing Assumption on Aerosol–Cloud Interactions in Marine Stratocumulus', J. Atmos. Sci., doi:10.1175/2008JAS2909.1

Hill et al., 2014, 'Mixed-phase clouds in a turbulent environment. Part 1: Large-eddy simulation experiments' Q.J.R. Meteorol. Soc., 140: 855-869. doi:10.1002/qj.2177

Koch and Del Genio, 2010, 'Black carbon semi-direct effects on cloud cover: review and synthesis', Atmos. Chem. Phys., doi:10.5194/acp-10-7685-2010

Ovchinnikov et al., 2014, 'Intercomparison of large-eddy simulations of Arctic mixed-phase clouds: Importance of ice size distribution assumptions', J. Adv. Model. Earth Syst., 6, 223– 248, doi:10.1002/2013MS000282.

Petters et al., 2013, 'A comparative study of the response of modeled non-drizzling stratocumulus to meteorological and aerosol perturbations', Atmos. Chem. Phys., doi:10.5194/acp-13-2507-2013

Wood, 2012, 'Stratocumulus Clouds', Mon. Weather Rev., doi:10.1175/MWR-D-11-00121.1

Yamaguchi et al., 2015, 'Stratocumulus to cumulus transition in the presence of elevated smoke layers', Geophys. Res. Lett., doi:10.1002/2015GL06654

Zhou et al., 2017, 'Impacts of solar-absorbing aerosol layers on the transition of stratocumulus to trade cumulus clouds', Atmos. Chem. Phys., doi:10.5194/acp-17-12725-2017

---

## Referee Comment (RC3) · Anonymous Referee #3 · 15 Oct 2019

**Review of "Diurnal cycle of the semi–direct effect over marine stratocumulus in large–eddy simulations" by Ross J. Herbert, Nicolas Bellouin, Ellie J. Highwood, Adrian A. Hill.**

The objective of this study is to investigate the role of an elevated aerosol layer on the evolution of a subjacent stratocumulus deck and the resulting changes in the overall radiative budget of the atmosphere integrated in the so-called the semi-direct effect (SDE). For this purpose a 3D small scale atmospheric model (LEM) was applied simulating the cloud evolution of a stratocumulus deck over a domain of 5.2x5.2 km$^2$ for a time period of 2 weeks, forcing the model by the diurnal cycle of SW radiation at 33°N during mid-July. After one-week period of "spin-up" a persistent aerosol layer was added above the cloud. The presence of the aerosol particles modifies the temperature conditions above the BL due to radiative heating by the absorbing aerosol during daytime. Consequently turbulent exchange processes (entrainment, detrainment) at the interface BL top / FT are altered (generally reduced) causing changes in the vertical structure of heat, turbulence and moisture over the entire BL.
The study investigates primarily the role of the depth of the aerosol layer, its distance from the cloud top and its aerosol optical depth. In this context the overall findings and conclusions are:
1) a persistent aerosol layer increases slightly SDE during daytime and
2) the strength of the SDE can vary considerably during daytime, what sheds doubt on the reliability of SDE data established from satellite observations only available during short daytime periods.
In addition also changes in the model set-up (i.e. modifications in large scale cooling, SST, FT moisture) and model processes (*no-rain* case) were imposed that confirm generally the previous results for SDE.
Only the increase of the BL depth up to 800 and 1000 m allows the suppression of a negative forcing due to the SDE.

All experiments investigate how daytime temperature perturbations just above the BL modify the BL's structure and its development. The cloud response is limited to the thermodynamic and turbulent changes in the BL and is thus only indirectly connected to the presence of an elevated aerosol layer.
*General critic*: the paper gives the impression that the general influence of absorbing aerosol on the cloud evolution has been investigated. However, it is only the radiative effect of the absorbing aerosol layer that enters in the discussion on the SDE, not the effect of aerosols "polluting" and thus modifying additionally the thermodynamic and hydrological cycle of the cloud and the BL. This should be highlighted in abstract and introduction, and should not only be mentioned at the end of the conclusion. The title of the paper should already include the key parameter of this study, i.e. " a persistent absorbing aerosol layer".
It is the LES model and its turbulence closure, which primarily determine the simulation results. Also the cloud description, even in simplified way, as it is in this study, affects the finding. Using other LES and cloud models (as done by other studies) will lead to a different result for the semi direct effect. This should be also highlighted in the conclusions of this paper.

*Although the study has also a couple of weaknesses (more details are below) the paper can be accepted for publication after corrections and improvements.*

Detailed remarks:

123   is the radiation code applied for all 130x130 columns individually or only for one mean profiles of T, Qvap and Qliq ?

212-220 makes it almost impossible to understand the calculation of the SDE. A reference would be

helpful (Hill and Dobbie, 1980?). I guess DRE uses the results of the simulations with the aerosol layer, actually not explained in the paper.

233   in caption of Fig. 3: w'w'w' is named the perturbation in mean vertical velocity. No, it is the perturbation of w to the power of 3, but it has another physical meaning. Why was it selected to illustrate the BL turbulence characteristics?

239-240   "During the daytime, ... through weakened surface to atmosphere gradients". The total water profiles (which are dominated by the presence $Q_{vap}$) for t = 13h and t = 5.30h in Fig.3c illustrate the contrary. Only the surface gradient in the first 10-20 m above the sea at 5.30h is stronger than the daytime conditions.
Also for THETA_liq in Fig. 3b a weak gradient exists during daytime but none at 5.30h.
This explanation of the decoupled state of the BL during daytime is not really convincing. Fig.3d and e better indicate the daytime / nighttime differences in the BL which are controlled by the vertical turbulent transport of tke and thus by the turbulence simulated in the LES model.

241   SST is kept constant, how are surface heat and moisture fluxes calculated? Give the key parameters.

274   Terminology in the caption of Fig.4: instead of "response" a more explicit description like "differences between no-aerosol simulation and simulation with an elevated aerosol layer" would make the illustration ∆Cloud, ∆LWP and ∆Albedo and the reference to equation (1) more understandable.

298   ... allows the cloud layer to maintain a higher RH.  This is difficult to understand and to believe, as explained in 2.2 the cloud model excess supersaturation is converted in liquid water.

301 -302 ... enhanced RH below cloud (caused by an increase in water vapour) ... by the decrease in latent heat flux. This is not credible. $Q_{total}$, i.e. mainly $Q_{vap}$, continuously decreases; also LHF mainly decreases but RH increases. RH is a function of two independent state variables T and $Q_{vap}$. What happened to the temperature T during the "aerosol" simulations in the BL. The paper completely omits a discussion on changes of the T profile in the BL after section 3.1. Temperature perturbations above the BL are the key parameter for the different SDE scenario in this study but a discussion of subsequent temperature changes in the BL is completely ignored  - why?

308   where can we see thicker clouds in Fig.5a in the afternoon of day 3? I can't.

335-336   "The decrease in cloud layer height allows better mixing beneath the cloud base, which enhances the evaporation of moisture from the surface between 0900 and 1500 ". This is in any case a speculation. It is not coherent with the daytime turbulence profile of Fig. 3e for the non-aerosol simulation.

346   rephrase "reduction in evaporation and associated cooling of entrained air". What do you mean with cooling of entrained air?

347   the reduced vertical motions reduce surface evaporation. Same question as above for 241.

394   Fig. caption 7: same remark as in 274

416   … quicker than the cloud base. Where or how to detect?

520   the *norain* simulation is not a model setup modification, but a change in the modeling physics.

522   the strengthening of the SDE is +1 W/m2 (or +3) from -7 to -8 W/m2 (or -9 to -12)
why not −1 W/m2 (and −3)?

527 − 535    This discussion or interpretation of the results cannot be understood with the given information on the *05cool* simulations. Thus, this part should be omitted.

537 and 547   Are the *SST* and *wetFT* simulations really with no-aerosol? This is probably a typo.

549   allowing the BL to maintain *below the cloud layer* a greater RH ?

564-565    … a redistribution of water from the cloud layer to the surface layer (Fig.9b). What do you mean with redistribution? Does it mean that rain is responsible for the significant Qvap increase in the first 300 m? This is most unlikely. Water vapor is accumulated in the lowest levels due to surface evaporation in the decoupled BL.

630    3b.  The conclusion that RH increase as Q_total increase, implicates that the BL temperature remains constant or decreases. This study, however, withholds this fundamental information.

721   regional climate models as HadGEM certainly treat surface fluxes

---

## Author Comment (AC3) · 12 Nov 2019

**Response to reviewer #3**

We thank the reviewer for their helpful comments. We begin by responding to the general comments, then we address each detailed remark in turn below. Reviewer comments are in bold, and changes made to the manuscript in italics.

General comments:

**The paper gives the impression that the general influence of absorbing aerosol on the cloud evolution has been investigated. However, it is only the radiative effect of the absorbing aerosol layer that enters in the discussion on the SDE, not the effect of aerosols "polluting" and thus modifying additionally the thermodynamic and hydrological cycle of the cloud and the BL. This should be highlighted in abstract and introduction, and should not only be mentioned at the end of the conclusion. The title of the paper should already include the key parameter of this study, i.e. " a persistent absorbing aerosol layer".**

As suggested by the reviewer, we have highlighted the focus of our study in the title, abstract, and introduction.

The title has been changed to *"Diurnal cycle of the semi–direct effect from a persistent absorbing aerosol layer over marine stratocumulus in large–eddy simulations"*

Our experiment description in the abstract has been extended and now reads: *"Here we use large eddy simulations to investigate the sensitivity of stratocumulus clouds to the properties of an absorbing aerosol layer located above the inversion layer, with a focus on the location, timing, and strength of the radiative heat perturbation"*

At the end of the introduction, when the outline of the study is summarised, we include the following: *"In this study the UK Met Office Large Eddy Model (LEM) is used to investigate and quantify the impact that the properties of an elevated absorbing aerosol layer have on the cloud and radiative response of marine stratocumulus, with a focus on the role that the location, timing, and strength of the heat perturbation has on the underlying cloud and boundary layer."*

**It is the LES model and its turbulence closure, which primarily determine the simulation results. Also the cloud description, even in simplified way, as it is in this study, affects the finding. Using other LES and cloud models (as done by other studies) will lead to a different result for the semi direct effect. This should be also highlighted in the conclusions of this paper.**

We agree that past literature has shown a large amount of diversity when different LES models are used to study semi-direct effects, although we expect that the fundamental causal chain explaining the response in our model would also be involved in the response simulated by other models. The following has been included in the conclusions section:

*"Inconsistent responses between LES models can also arise through differences in the representation of processes, including unresolved sub-grid scale turbulence (e.g., Stevens et al. 2005) and microphysics (van der Dussen et al., 2013). Our results show that the heat perturbation above the cloud layer impacts all aspects of the BL profile, therefore it would be beneficial to repeat this study using other LES models to test our conclusions."*

**123 is the radiation code applied for all 130x130 columns individually or only for one mean profiles of T, Qvap and Qliq?**

The radiation scheme is applied to all columns. We have clarified this in Section 2.2 with the following text:

"Radiation calculations are performed for all grid points within the domain every 30 seconds."

**212-220 makes it almost impossible to understand the calculation of the SDE. A reference would be helpful (Hill and Dobbie, 1980?). I guess DRE uses the results of the simulations with the aerosol layer, actually not explained in the paper.**

As suggested, we have included a reference to Johnson et al. 2004 which provides a description of the SDE calculation.

We state how the DRE is calculated in lines 250 to 251, but have added a sentence to clarify:

*"DRE is calculated as the difference between $F_{TOA}$ and that obtained in a second, diagnostic, call to the radiation scheme with the same profiles of liquid water, water vapour, and atmospheric gases, but without aerosol. This second call is only performed for the simulations with aerosol present."*

**233 in caption of Fig. 3: w'w'w' is named the perturbation in mean vertical velocity. No, it is the perturbation of w to the power of 3, but it has another physical meaning. Why was it selected to illustrate the BL turbulence characteristics?**

We agree that a more appropriate variable to illustrate BL turbulence is the mean variance in vertical velocity perturbation ($\overline{w'w'}$). We have updated figure 3 and relevant text in Section 3.1, which is the only instance of use throughout the manuscript.

**239-240 "During the daytime, … through weakened surface to atmosphere gradients". The total water profiles (which are dominated by the presence Q_vap) for t = 13h and t = 5.30h in Fig.3c illustrate the contrary. Only the surface gradient in the first 10-20 m above the sea at 5.30h is stronger than the daytime conditions. Also for THETA_liq in Fig. 3b a weak gradient exists during daytime but none at 5.30h. This explanation of the decoupled state of the BL during daytime is not really convincing. Fig.3d and e better indicate the daytime / nighttime differences in the BL which are controlled by the vertical turbulent transport of tke and thus by the turbulence simulated in the LES model.**

We agree with the reviewer that our explanation did not provide a convincing description of the day / night differences in BL turbulence. As suggested, we have used figures 3d and 3e to illustrate the different turbulent structures and the decoupling during daytime. The sentence now reads:

*"During the daytime, solar heating reduces the buoyancy flux (Fig. 3d) through an offset in the longwave cooling and reduces turbulence throughout the BL (Fig. 3e). This weakens the BL circulation and prevents mixing throughout the BL and promotes a decoupled state in which the flux of moisture from the surface to the cloud is insufficient to maintain the cloud base height, as evident from the non–constant BL profiles of $\theta_l$ (Fig. 3b) and $q_t$ (Fig. 3c) at 1300 hours."*

**241 SST is kept constant, how are surface heat and moisture fluxes calculated? Give the key parameters.**

The surface fluxes are calculated using Monin–Obukhov similarity theory, which is reported in section 2.1 (Description of model; line 117). As suggested by the reviewer, we have expanded on this to provide more information. The sentence now reads:

*"Surface fluxes of moisture and heat are calculated using Monin–Obukhov similarity theory (Monin and Obukhov, 1954) which predicts the surface frictional stresses and heat fluxes using the local gradients between the surface and the overlying model level. For these experiments a prescribed constant sea surface temperature is used."*

**274 Terminology in the caption of Fig.4: instead of "response" a more explicit description like "differences between no-aerosol simulation and simulation with an elevated aerosol layer" would make the illustration DCloud, DLWP and DAlbedo and the reference to equation (1) more understandable.**

The figure caption has been amended as suggested.

**298 … allows the cloud layer to maintain a higher RH. This is difficult to understand and to believe, as explained in 2.2 the cloud model excess supersaturation is converted in liquid water.**

The reviewer is correct. We erroneously referred to the cloud layer, rather than the sub-cloud layer. This has been corrected to read:

*"The increase in RH occurs due to the weakened $w_e$ which reduces the amount of warm dry FT air that is mixed into the BL and allows the sub–cloud layer to maintain a higher RH."*

**301 -302 … enhanced RH below cloud (caused by an increase in water vapour) … by the decrease in latent heat flux. This is not credible. Q_total, i.e. mainly Q_vap, continuously decreases; also LHF mainly decreases but RH increases. RH is a function of two independent state variables T and Q_vap. What happened to the temperature T during the "aerosol" simulations in the BL.**

The RH below cloud increases (Fig. 5f) due to the increase in qt (total water specific humidity) below cloud, which occurs despite a decrease in the total water path (Fig. 5f). We believe we introduced some confusion with the variable name 'total BL qt' which, as pointed out by reviewer #2, should be correctly termed total water path.

We chose not to show the qt below the cloud in these plots, and instead use the decrease in LHF from the surface and increase in mean RH below cloud to demonstrate this response,

which occurs alongside a small decrease in BL liquid-water potential temperature of ~0.1 K (discussed in more detail below). The increase in mean qt can be seen further on in the manuscript in Fig. 10e.

We have addressed this by replacing 'total BL qt' with total water path (TWP) throughout the manuscript, and as discussed below we have introduced the mean BL liquid water potential temperature into Figures 5 and 6 to support our analysis.

**The paper completely omits a discussion on changes of the T profile in the BL after section 3.1. Temperature perturbations above the BL are the key parameter for the different SDE scenario in this study but a discussion of subsequent temperature changes in the BL is completely ignored - why?**

In our simulations the temperature only plays a minor role in the BL response, yet as correctly pointed out by the reviewer, we do not discuss this at any point. To address this, we now include the BL liquid-water potential temperature ($\theta_l$) response in Figures 5 and 6, which are shown below as Fig. R1 and R2, respectively.

The figures show that the mean $\theta_l$ decreases by ~0.1 K over the initial three days of the simulation (Fig. R1) and ~0.2 K after 10 days (Fig. R2). The sources for changes to the $\theta_l$ occur through exchanges across the inversion layer at the top of the BL, changes in LW and SW fluxes, surface fluxes, and precipitation. In our simulations the enhanced inversion strength reduces the flux of warm air into the BL, however, this may be partially offset by the heat perturbation produced by the absorbing aerosol layer. Small decreases in precipitation are offset by reduced latent heat flux at the surface, and due to the high cloud fraction in the simulations, changes to LW and SW fluxes within the BL are small. The pattern of the $\theta_l$ response in Fig. R1 is similar to the entrainment rate (Fig. 5b) which makes it likely that the process controlling the simulated response in $\theta_l$ is the change to the entrainment rate.

The magnitude of the response in both the initial and the steady-state response is small (up to 0.2 K in the steady-state) which demonstrates that the dominant driver of changes to RH below cloud is from the water vapour, rather than the temperature.

In addition to the new subplots in figures 5 and 6, we have included text in section 3.2.1:

*"The increase in RH occurs due to the weakened $w_e$ which reduces the amount of warm dry FT air that is mixed into the BL and allows the sub–cloud layer to maintain a higher RH. The relatively small decrease in potential temperature of ~ 0.1 K (Fig. 5g) suggests that the RH response is driven by an increase in water vapour."*

in section 3.2.2:

*"The small response in mean BL potential temperature of -0.2 K (Fig. 6f) strengthens the hypothesis that the RH response below-cloud is driven by changes in available water vapour, rather than the decrease in temperature, although it is worth noting that this decrease in temperature will act to slightly increase the RH."*

in section 4:

*"The reduction in the entrainment of warm and dry air from the FT reduces the amount of mixing, reducing the sink of $\overline{q_t}$ in the cloud layer and allowing the BL to maintain a greater RH. The result is an increase in $\overline{q_t}$, a small decrease in BL temperature, and an increase in RH."*

[Figure]

*Figure R1. 'Initial' domain averaged cloud response of BL liquid-water potential temperature – subplot taken from Figure 5 in the revised manuscript.*

[Figure]

*Figure R2. Domain averaged 'steady-state' cloud response of BL liquid-water potential temperature – subplot taken from Figure 6 in the revised manuscript.*

**308 where can we see thicker clouds in Fig.5a in the afternoon of day 3? I can't.**

The paragraph in question refers to the increase in LWP of ~ 2 g kg⁻¹ that is evident in Fig. 5c from just before midday and continuing through the afternoon on day 3. There is also a corresponding increase in the geometric thickness of the cloud at midday evident in Fig. 5a which shows that the cloud base has decreased more than the cloud top.

We have clarified the paragraph by starting the paragraph with the following:

*"The thicker cloud (enhanced LWP; Fig. 5a) on the afternoon of the third day…"*

A similar change has been made to the previous paragraph which discusses the thinner cloud on the morning of the third day.

**335-336 "The decrease in cloud layer height allows better mixing beneath the cloud base, which enhances the evaporation of moisture from the surface between 0900 and 1500 ". This is in any case a speculation. It is not coherent with the daytime turbulence profile of Fig. 3e for the non-aerosol simulation.**

A given eddy generated in the cloud layer will be able to penetrate a certain distance below the cloud. If the cloud layer is reduced in altitude then the eddy will be able to influence layers closer to the surface. We see evidence of this occurring in our simulations. Figure R3a shows that although the maximum turbulence in the cloud layer is slightly weaker in the aerosol simulation, the decrease in cloud altitude by ~ 200 m systematically shifts the w'w' profile downwards, so that turbulence below cloud is greater in the simulations with aerosol. The buoyancy flux profile (Fig. R3b) shows that this results in positive buoyancy flux below

cloud, as opposed to a negative value in the no-aerosol simulation, resulting in the response that we observe in Figure 6.

Figure R3 has been included in the supplementary information (as Figure S2) and is now referred to in the manuscript in section 3.2.2:

*"This modification to the diurnal cycle of the cloud is driven by an increased coupling between surface moisture flux and cloud base during the daytime (see Fig. S2 in the supplementary information) …"*

[Figure]

Figure R3. Domain-mean vertical profiles of a) variance in vertical velocity perturbation w'w', and b) buoyancy flux on day 13 of the simulation at 1300 local time for the no-aerosol simulation (black dashed line) and following the introduction of a layer of absorbing aerosol (blue solid line) in the base experiment (0 m cloud–aerosol gap, 250 m thick layer, and AOD of 0.2).

**346 rephrase "reduction in evaporation and associated cooling of entrained air". What do you mean with cooling of entrained air?**

The air that is entrained into the cloud layer from the free troposphere is dry. The subsequent mixing of the dry and cloudy air results in net evaporation which acts to cool the parcel of air, generating (negative) buoyancy.

As suggested, we have rephrased the sentence, which now reads:

*"The weakened BL circulation is therefore due to a reduction in entrainment. The mixing of dry air into the cloud layer results in evaporation and a cooling which generates buoyancy; a reduction in entrainment therefore weakens cloud–top buoyancy production."*

**347 the reduced vertical motions reduce surface evaporation. Same question as above for 241.**

As per the previous question the text concerning the surface fluxes in section 2.1 has been expanded to provide more information.

**394 Fig. caption 7: same remark as in 274.**

The caption has been changed as suggested.

**416 … quicker than the cloud base. Where or how to detect?**

The current manuscript does not provide evidence of this, as also highlighted by Reviewer #2. We have therefore included additional plots in the supplementary information and have rewritten the first paragraph of section 3.3.1 to discuss this in more detail. The paragraph reads:

*"The majority of experiments show a positive spike in SDE (Fig. 7d, i and n) just before midday on the first day. This occurs due to the lag–time in response between the direct impact to the cloud from changes to we, and the increase in sub–cloud RH. Figure S1 in the supplementary information focuses on the response in the initial 24 hours. The positive SDE is driven by the decrease in LWP caused by an increase in cloud base height (Fig. 5a and Fig. S1a) without a corresponding change in cloud top height. The decrease in $w_e$ weakens buoyancy production throughout the cloud layer (Fig. S1c), which drives a reduced moisture flux within the cloud and to the cloud base (Fig. S1d). As the day progresses the continued reduction of $w_e$ results in an increase in mean below–cloud RH and a recovery, or increase, of the LWP. This explains why stronger perturbations to the entrainment rate on the first day (such as when the layer is close to the cloud) results in a quicker recovery of the LWP (Fig. 7c, h, and m)."*

**520 the norain simulation is not a model setup modification, but a change in the modeling physics.**

Granted, although for the sake of simplicity we refer to those sensitivity experiments as experimenting with "model setup".

**522 the strengthening of the SDE is +1 W/m2 (or +3) from -7 to -8 W/m2 (or -9 to -12) why not -1 W/m2 (and -3)?**

We agree that the current text is confusing. We have rewritten the sentence to read:

*"Compared to the control setup the noRain setup is characterised by a consistent increase in the magnitude of the SDE by 1 $Wm^{-2}$ when a cloud–aerosol gap is present and up to 3 $Wm^{-2}$ when there is no gap"*

**527 – 535 This discussion or interpretation of the results cannot be understood with the given information on the 05cool simulations. Thus, this part should be omitted.**

As discussed in Section 3.4 the large-scale advective heat tendency is a large-scale forcing term that represents a degree of variability in LES experiments. We therefore do not want to omit this result, and have instead rewritten the paragraph to improve our interpretation of the results. The paragraph now reads:

*"When compared to the control setup, increasing the cooling rate of the large–scale advective heat tendency results in stronger BL dynamics, enhanced cloud–top entrainment*

*of warm dry air, and enhanced surface LHF (which acts as a feedback to enhanced entrainment). As the processes maintaining the cloud layer become more important, they become more sensitive to perturbations. Therefore, when the aerosol layer is present in the 05cool setup, the responses of $w_e$, LHF, and below–cloud moisture flux are stronger than in the control setup and the simulations are characterised by a quicker decrease in the TWP of the BL. However, this only becomes prominent on the third day and results in little difference from the control setup over the first two days."*

**537 and 547 Are the SST and wetFT simulations really with no-aerosol? This is probably a typo.**

This is not typo. In each set of sensitivity experiments a simulation was run with and without aerosols. This allows us to investigate the response of each setup to the same aerosol perturbation. For each setup we briefly describe what is changing in the 'without aerosol' simulations before discussing the response to the aerosol layers.

We have made this clearer in the revised manuscript by adding a sentence in Sect. 3.4 pointing out that a no–aerosol simulation is run for each setup and referring to the no–aerosol simulations within our interpretation.

**549 allowing the BL to maintain below the cloud layer a greater RH?**

This has been changed to:

*"…allowing the BL to maintain a greater mean RH"*

**564-565 ... a redistribution of water from the cloud layer to the surface layer (Fig.9b). What do you mean with redistribution? Does it mean that rain is responsible for the significant Qvap increase in the first 300 m? This is most unlikely. Water vapor is accumulated in the lowest levels due to surface evaporation in the decoupled BL.**

Yes, the reduced coupling between surface and cloud layers results in an accumulation of water vapor towards the surface, which can be viewed as a redistribution of the water. This does not occur due to precipitation processes.

The sentence has been amended to clarify our point:

*"This reduces the flux of water vapour from the surface layer to the cloud, resulting in an accumulation of water vapour close to the surface (Fig. 9b)."*

**630 3b. The conclusion that RH increase as Q_total increase, implicates that the BL temperature remains constant or decreases. This study, however, withholds this fundamental information.**

As per our previous discussion on the temperature response, this line has been amended to read:

*"The reduction in the entrainment of warm and dry air from the FT reduces the amount of mixing, reducing the sink of $\overline{q_t}$ in the cloud layer and allowing the BL to maintain a greater*

*RH. The result is an increase in $\bar{q}_t$, a small decrease in BL temperature, and an increase in RH."*

**721 regional climate models as HadGEM certainly treat surface fluxes**

We have amended the sentence as follows:

*"The lack of BL adjustment may be due to processes that are not explicitly treated in HadGEM, such as BL turbulence and subsequent missing feedbacks on surface fluxes…"*

**References**

Stevens, B., and Coauthors, 2005: Evaluation of large-eddy simulations via observations of nocturnal marine stratocumulus. Mon. Wea. Rev., 133, 1443–1462, https://doi.org/10.1175/MWR2930.1.

van der Dussen, J. J., and Coauthors, 2013: The GASS/EUCLIPSE model intercomparison of the stratocumulus transition as observed during ASTEX: LES results, J. Adv. Model. Earth Syst., 5, 483–499, doi:10.1002/jame.20033.